# Inhibition of DNA methylation promotes breast tumor sensitivity to netrin-1 interference

Mélodie Grandin[1], Pauline Mathot[1], Guillaume Devailly[1], Yannick Bidet[2], Akram Ghantous[3], Clementine Favrot[1], Benjamin Gibert[1], Nicolas Gadot[4], Isabelle Puisieux[5], Zdenko Herceg[3], Jean-Guy Delcros[1], Agnès Bernet[1], Patrick Mehlen[1,*,†] & Robert Dante[1,**,†]

## Abstract

In a number of human cancers, NTN1 upregulation inhibits apoptosis induced by its so-called dependence receptors DCC and UNC5H, thus promoting tumor progression. In other cancers however, the selective inhibition of this dependence receptor death pathway relies on the silencing of pro-apoptotic effector proteins. We show here that a substantial fraction of human breast tumors exhibits simultaneous DNA methylation-dependent loss of expression of NTN1 and of DAPK1, a serine threonine kinase known to transduce the netrin-1 dependence receptor pro-apoptotic pathway. The inhibition of DNA methylation by drugs such as decitabine restores the expression of both NTN1 and DAPK1 in netrin-1-low cancer cells. Furthermore, a combination of decitabine with NTN1 silencing strategies or with an anti-netrin-1 neutralizing antibody potentiates tumor cell death and efficiently blocks tumor growth in different animal models. Thus, combining DNA methylation inhibitors with netrin-1 neutralizing agents may be a valuable strategy for combating cancer.

**Keywords** apoptosis; breast cancer; decitabine; dependence receptor; DNA methylation

**Subject Categories** Cancer; Chromatin, Epigenetics, Genomics & Functional Genomics

## Introduction

Recent research focusing on a specific functional family of receptors, namely the dependence receptors (DRs) (Mehlen *et al*, 1998; Llambi *et al*, 2001), has highlighted their implication in inhibiting tumor progression. Indeed, in contrast to most cellular receptors, a dual role characterizes these transmembrane receptors: in the presence of their respective ligands, they provide a typical positive signal (promoting cell survival, migration, proliferation, etc.), while the absence of ligand triggers a cascade of signaling events leading to apoptotic cell death. The DR protein family currently contains approximately twenty known members, and one of their most exhaustively studied ligands is Netrin-1. This DR ligand was the first secreted attractive axon guidance cue to be described in the early nineties, since then, the role of netrin-1 in many biological functions has been established (Cirulli & Yebra, 2007; Mehlen & Guenebeaud, 2010; Ramkhelawon *et al*, 2014). Netrin-1 receptors, namely, are the Deleted in Colorectal Carcinoma (DCC), the Uncoordinated-5-Homologs (UNC5H1-4/A–D). The ability of these receptors to trigger apoptosis in settings under ligand-limited conditions was shown to be a constrain for impede tumor progression. Indeed, the inactivation of *UNC5C*, and *DCC*, or the mutation of the *DCC*-inducing apoptosis domain is associated with tumor development and progression in mouse models (Bernet *et al*, 2007; Castets *et al*, 2012; Krimpenfort *et al*, 2012). Furthermore, in line with these observations, *DCC* and *UNC5H* are silenced in many cancers, either by loss of heterozygosity or epigenetic mechanisms (Hedrick *et al*, 1994; Bernet *et al*, 2007; Shin *et al*, 2007). Alternatively, in some types of cancers, an upregulation of *NTN1* provides a similar tumor growth selective advantage by abolishing their dependency on netrin-1 availability in the micro-environment (Fitamant *et al*, 2008). This upregulation is particularly marked in some cases of aggressive breast cancer and in breast metastasis (Fitamant *et al*, 2008; Harter *et al*, 2014). The reliance on netrin-1 for tumor growth was rapidly perceived as an opportunity for therapeutic intervention, since it was speculated that the disruption of the binding of netrin-1 to its receptors should induce apoptotic cell death *in vitro* and tumor growth inhibition *in vivo*. Consistently, the silencing of netrin-1 or the development of biological agents interfering with netrin-1/DRs interactions has been

1. Dependence Receptors, Cancer and Development Laboratory - Equipe labellisée 'La Ligue', LabEx DEVweCAN, Centre de Recherche en Cancérologie de Lyon (CRCL), INSERM U1052-CNRS UMR5286, Université de Lyon, Centre Léon Bérard, Lyon, France
2. Laboratoire d'Oncologie Moléculaire, Centre Jean Perrin, Clermont-Ferrand, France
3. Epigenetics Group, IARC, Lyon, France
4. Endocrine Differentiation Laboratory, CRCL, Université de Lyon, Hospices Civils de Lyon, Hôpital Edouard Herriot, Anatomie Pathologique, Lyon, France
5. Targeting of the tumor and its immune environment Laboratory CRCL, INSERM U1052, CNRS UMR5286, UCBL, CLB, Lyon, France
   *Corresponding author. Tel: +33 4 7878 2870; E-mail: patrick.mehlen@lyon.unicancer.fr
   **Corresponding author. Tel: +33 7 7878 5922; E-mail: robert.dante@lyon.unicancer.fr
   †These authors contributed equally to this work

shown to efficiently reduce tumor growth and metastasis in different animal models (Fitamant et al, 2008; Delloye-Bourgeois et al, 2009a,b; Paradisi et al, 2013; Grandin et al, 2016). Moreover, an anti-netrin-1 antibody is under preclinical evaluation and should be assessed in early clinical trials in 2016.

Nevertheless, a substantial fraction of human tumors appears to conserve the expression of netrin-1 receptors without upregulating NTN1 expression, suggesting that the downstream DR pathways may be impaired (Shin et al, 2007; Mian et al, 2011; Krimpenfort et al, 2012). In cancers, epigenetic modifications are frequently associated with an increase in the expression of anti-apoptotic proteins and with the inactivation of factors inducing apoptosis (Baylin & Ohm, 2006). In this context, previous studies have reported that DAPK1, a serine threonine kinase responsible for UNC5H-induced apoptosis, is downregulated in various cancers (Llambi et al, 2005; Guenebeaud et al, 2010). Furthermore, mechanistic analyses have demonstrated a direct relationship between the hypermethylation of the CpG island (CGi) located within the DAPK1 promoter region and its downregulation (Raval et al, 2007; Pulling et al, 2009; Mian et al, 2011; Kilinc et al, 2012). Finally, it was shown that treatment with decitabine (5-aza-2′-deoxycytidine, DAC) inhibits the DNA methylation of the DAPK1 promoter and restores DAPK1 expression in lung cancer cell lines (Tang et al, 2004).

Inhibition of DNA methylation may prove to be a promising approach for combating cancer (Yang et al, 2010; Rodriguez-Paredes & Esteller, 2011a; Dawson & Kouzarides, 2012; Tsai et al, 2012). Concordantly, DAC showed anti-leukemic effects in several clinical trials and has been approved by FDA for the treatment of myelodysplastic syndromes and recent studies indicated that DAC shows some effects in preclinical models (Tsai et al, 2012). Here, we report that treatment with DAC restores the pro-apoptotic machinery linked to the netrin-1/DR-mediated cell death signaling pathway in breast cancer cell lines and patient-derived xenografts and sensitize cancer cells to netrin-1 neutralizing agents. We thus provide evidence that combining DNA methylation inhibitors with a netrin-1 neutralizing antibody empowers tumor cell death in vitro and tumor growth inhibition in mice.

## Results

### Epigenetic downregulation of netrin-1 is associated with the epigenetic downregulation of DAPK1, in human breast cancers

In order to study the mechanisms underlying inhibition of netrin-1/DRs-mediated apoptosis, which in a fraction of tumors is neither dependent on netrin-1 upregulation nor DR silencing, we investigated DAPK1 expression and DNA methylation in breast tumors.

We thus examined differentially methylated regions (DMRs) associated with malignant transformation in breast cancer samples ($n = 92$) available from The Cancer Genome Atlas (TCGA) by comparing paired normal/tumoral data obtained from the HumanMethylation450K Array (HM450, Illumina) and by high-throughput sequencing of polyadenylated RNA (RNAseq). This comparison focused on the level of DNA methylation within the 5′-end of NTN1 (position −765 to + 1,300 relative to the transcription start site, TSS) and DAPK1 (+ 365 and + 838 relative to the TSS) and revealed that these regions were hypermethylated (threshold = 2) in

about 30% of tumoral samples compared with their normal counterparts (Fig 1A and B). Furthermore, NTN1 was downregulated (Fig 1C, fold change (FC) ≤ 0.5) in 43% of cases and the "NTN1-low" samples were hypermethylated (Fig 1G; $P = 3 \times 10^{-2}$, two-sided Mann–Whitney test) when compared with samples exhibiting no NTN1 downregulation (FC ≥ 1.3). Using the same approach for DAPK1, we observed that samples exhibiting DAPK1 downregulation (29% of the samples, Fig 1D) were also hypermethylated ($P = 3 \times 10^{-4}$) when compared with the other samples (Fig 1H). These epigenetic modifications did not seem to be independent events since in NTN1-hypermethylated samples ($n = 23$, FC ≥ 2), DAPK1 was also hypermethylated (mean FC = 2.22), while in NTN1-hypomethylated samples ($n = 13$, FC < 0.7), DAPK1 was not hypermethylated (mean FC = 0.93). The relationship between DAPK1/NTN1 downregulation and DNA hypermethylation was also observed in a larger number of breast cancer samples ($n = 807$) available on TCGA data portal. Indeed, the mean percentage of CpG methylation (Fig 1E and F) of DAPK1 and NTN1 were inversely correlated with their levels of expression (Pearson's $r = -0.32$, $P < 10^{-18}$ and Pearson's $r = -0.14$, $P = 6.7 \times 10^{-5}$, respectively), suggesting that DNA methylation represses DAPK1 and NTN1 transcription in human breast tumors.

Although we cannot excluded a bias due to stromal contamination, it should be noted that, in paired samples, while hypermethylation was observed in tumoral tissues, very low levels of DNA methylation were measured in their normal counterparts (Fig 1A and B). For both genes, a region (not represented on HM450) located at the 3′-end of the CGis (Fig 2A, light gray boxes) was selected for the quantitative analysis of DNA methylation of breast cancer biopsies (tumor bank—Centre Léon Bérard) by bisulfite pyrosequencing. The analysis indicated that the mean percentage of CpG methylation of the DAPK1 and NTN1 pyrosequenced regions were inversely correlated (Pearson's $r = -0.66$, $P = 0.003$, and Pearson's $r = -0.55$, $P = 0.008$, respectively) with their levels of expression (Fig EV1A and B). Altogether, these data suggested that DNA methylation is involved in the downregulation of NTN1 and DAPK1 in human breast cancers.

To determine whether this concomitant change in DAPK1 and NTN1 expression was also observed at the protein levels, DAPK1, UNC5B, and netrin-1 were measured by immunohistochemistry (IHC) using tissue microarrays (70 sections) from human breast ductal carcinoma (Super Bio Chips). This analysis revealed, that, netrin-1-low samples also exhibited low levels of DAPK1 ($\chi^2$ test, $P = 0.04$). In contrast, UNC5B levels were the same, irrespective of netrin-1 levels (Fig EV1C and D). This result was validated in the TCGA cohort of breast tumors samples ($n = 807$), where a correlation between DAPK1 and NTN1 expression was observed (odd ratio = 4.71, $P = 0.03$, Fig EV1E).

In order to establish an in vitro experimental model mimicking the DNA methylation alterations observed in breast cancer tissues, DNA methylation patterns of two cancer cell lines were determined by parallel sequencing. Pull-down assays were conducted using the MDA-MB-231 cell line derived from human breast cancer, and the HMLER cell line generated through the in vitro transformation of human mammary cells (Elenbaas et al, 2001; Morel et al, 2008). Methylated DNA fragments were selected using a recombinant protein containing the Methyl-CpG-binding domain of MBD2 of MBD2 (Methyl-Cap-seq), from MDA-MB-231 cell line derived from

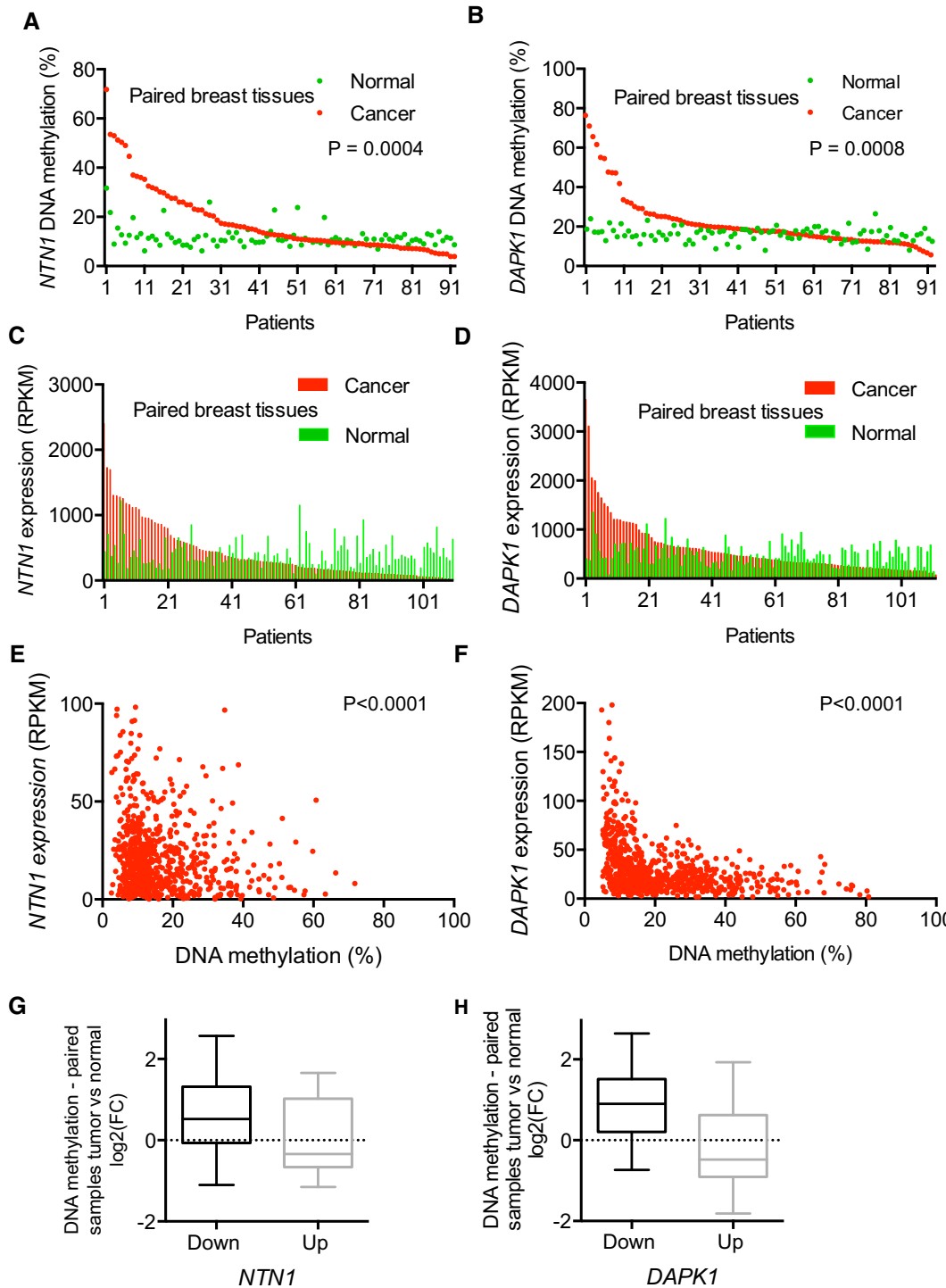

**Figure 1.  *NTN1* and *DAPK1* are hypermethylated and downregulated in human breast cancers.**

A, B     DNA methylation level of *NTN1* (A) and *DAPK1* (B) 5′ regions (Illumina's HumanMethylation450K Array (HM450) from The Cancer Genome Atlas breast cohort) in paired breast tissues (normal: green circles, tumor: red circles), *n* = 92, Wilcoxon matched-pairs signed rank test, *P* = 0.004 and *P* = 0.0008.

C, D     *NTN1* (D) and *DAPK1* (G) gene expression in paired breast tissues (normal: green bars, tumor: red bars), RNAseq from TCGA breast cohort, *n* = 112 and 114, respectively.

E, F     Correlation between *NTN1* (E) and *DAPK1* (F) gene expression and DNA methylation in the breast cancer cohort (TCGA, *n* = 807). Pearson correlation, $P = 6.7.10^{-5}$, $r = -0.14$ for *NTN1* (A) and $P < 10^{-16}$, $r = -0.32$ for *DAPK1* (B), respectively.

G, H     Tumor/normal DNA methylation ratio of *NTN1* (G) and *DAPK1* (H) in human breast tumors (data extracted from TCGA cohort, paired samples) according to gene expression (downregulated FC ≤ 0.5, down, *n* = 33, or upregulated FC ≥ 1.3, up, *n* = 16). $P = 3 \times 10^{-2}$ and $P = 3 \times 10^{-4}$ two-sided Mann–Whitney test, for *NTN1* and *DAPK1*, respectively.

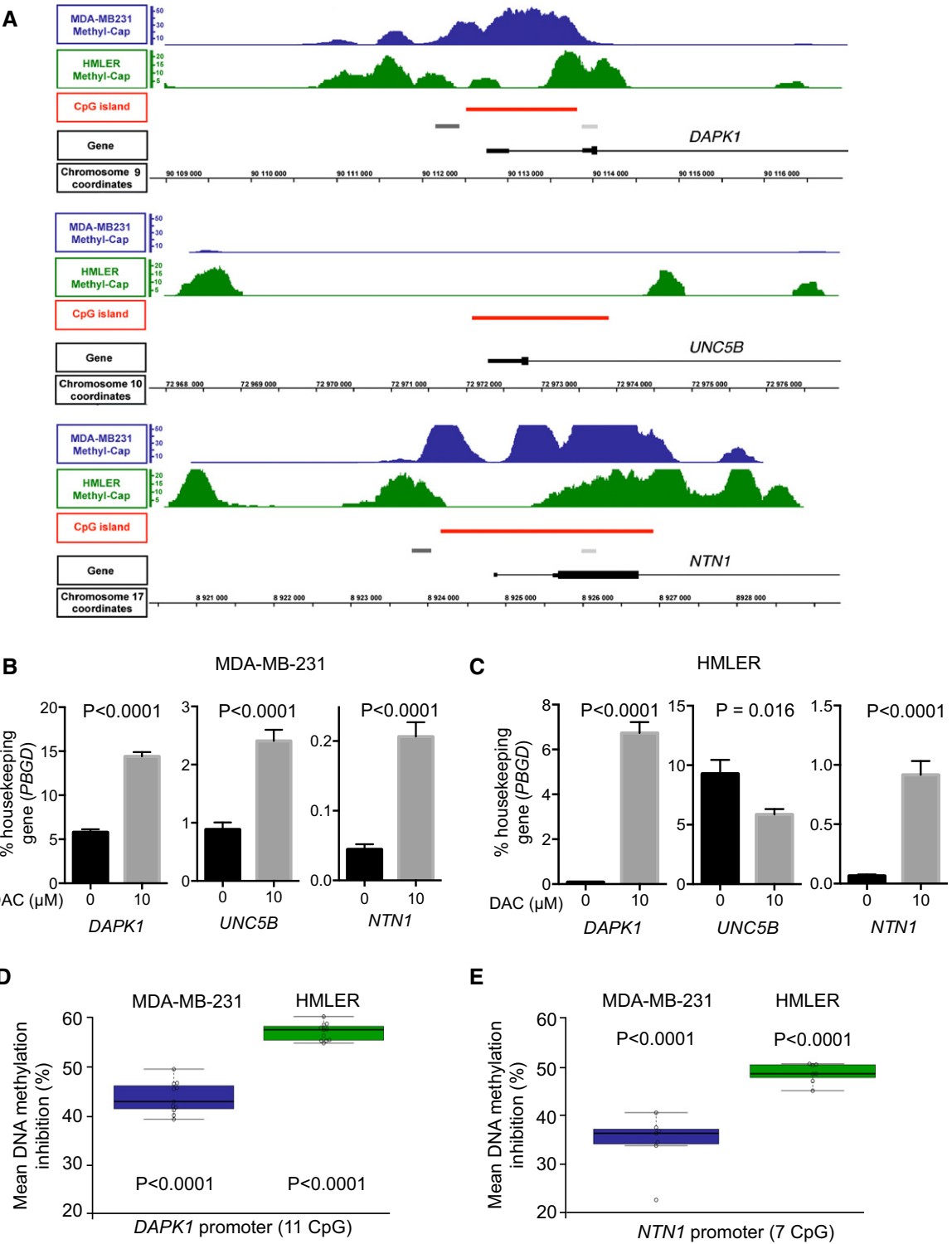

**Figure 2.  DNA methylation and demethylation in mammary cell lines.**

A    Methyl-Cap-seq read density profiles of the 5′ end of *DAPK1*, *UNC5B*, and *NTN1* in MDA-MB-231 (blue) and HMLER (green) cells. Red boxes represent the CpG islands (CGIs); light gray boxes the regions analyzed by bisulfite PCR sequencing; dark gray boxes represent the regions analyzed by parallel sequencing of amplicons from bisulfite modified DNA; and black boxes represent the exons and UTR. Chromosome coordinates of each gene are given (black lines).

B, C    Gene expression was measured by qRT–PCR after 72 h for MDA-MB-231 (B) and HMLER cells (C) treated daily with DAC (10 μM). *PBGD* expression level was used as an internal control. Data are expressed as mean ± s.e.m. of at least 3 independent experiments. ****P < 0.0001, two-tailed unpaired Student's *t*-test.

D, E    Measurement of DNA hypomethylation of the *DAPK1* (D) and *NTN1* (E) promoters after decitabine treatment of MDA-MB-231 and HMLER cells. Over 1960 sequences were analyzed per group in 2 independent experiments. ****P < 0.0001, two-way ANOVA and *post hoc* Tukey test.

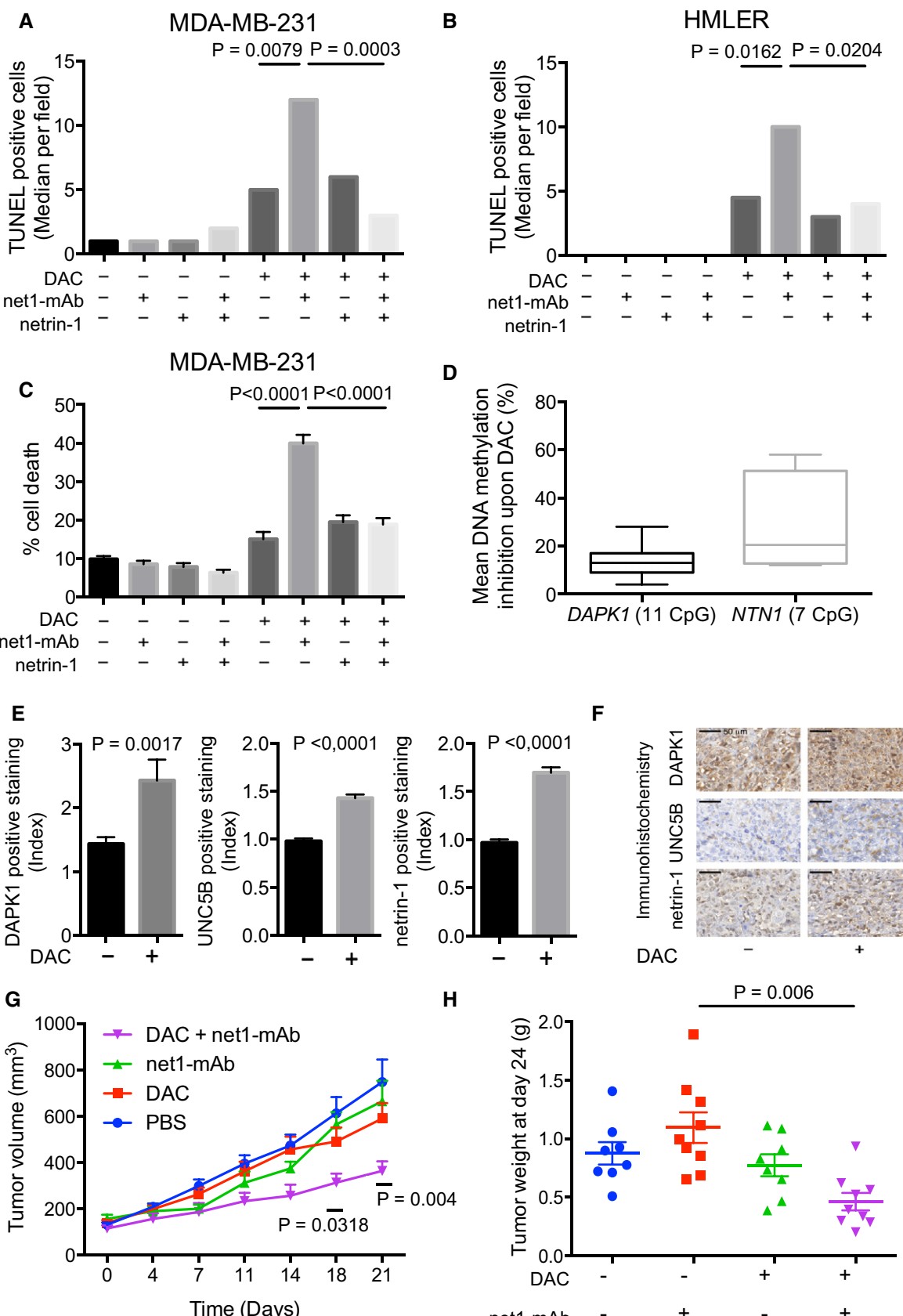

Figure 3.

**Figure 3.  Netrin-1 neutralizing antibody net1-mAb triggers apoptosis in hypomethylated MDA-MB-231 and HMLER breast cancer cell lines.**

A, B  Cells were treated with DAC (10 µM, 72 h), and/or net1-mAb (10 µg/ml, 48 h) and/or recombinant netrin-1 (5 µg/ml, 48 h). TUNEL assays from 3 independent experiments. \*\*\*$P < 0.0001$ one-way ANOVA.

C  Cellular mortality was assessed using the toxilight assay. Data represent mean ± s.e.m. from 3 independent experiments. \*\*\*\*$P < 0.0001$, one-way ANOVA.

D–H  MDA-MB-231 cells were implanted into the mammary gland fat pad of immuno-compromised mice. When tumors reached 100 mm³, mice were injected subcutaneously with decitabine (0.4 mg/kg) or PBS and/or intraperitoneally with net1-mAb (10 mg/kg). (D) Loss of DNA methylation of the *NTN1* and *DAPK1* promoters in decitabine treated tumors, compared with PBS treated tumors, after one week of treatment. > 1,700 sequences were analyzed per group in 2 independent experiments. \*\*\*\*$P < 0.0001$, two-way ANOVA and *post hoc* Tukey-test. (E, F) After one week of treatment, tumors from xenografted mice were fixed in formalin, embedded in paraffin, and sliced into 4 µm sections. (E) Levels of DAPK1, UNC5B, and netrin-1 were measured in 4 independent tumors per treatment group by immunohistochemistry staining, and expressed as a percentage of total tumor surface normalized against the control PBS-group. Data represented as means ± s.e.m. \*\*\*\*$P < 0.0001$, two-way ANOVA and *post hoc* Tukey-test. (F) Representative tumor sections corresponding to MDA-MB-231 xenografts, scale bars = 50 µm. (G) Tumor volumes were measured twice a week. The statistical significance of the differences obtained between the control PBS-group and treated (DAC + net1-mAb) group was determined by two-way ANOVA and *post hoc* Tukey-test. \*\*\*$P < 0.0001$. Error bars = s.e.m. $n = 9$ mice per group. (H) Tumor weights were measured 3 days after the end of the experiment. The statistical significance of the differences obtained between the groups was determined by one-way ANOVA, \*\*\*$P < 0.0001$. Error bars = s.e.m. $n = 9$ mice per group.

human breast cancer, and the HMLER cell line constructed from *in vitro* transformation of human mammary cells (Elenbaas *et al*, 2001; Morel *et al*, 2008). Data obtained indicated that the 5′-end CpGis of *DAPK1* and *NTN1* were methylated in the two cancer cell lines (Fig 2A). Furthermore, the treatment of MDA-MB-231 and HMLER cells with decitabine (DAC) resulted in a significant upregulation of *DAPK1* and *NTN1* mRNA (Fig 2B and C) and in the inhibition of DNA methylation within the *DAPK1* and *NTN1* promoter regions (Figs 2D and E, and EV1F and G).

Parallel sequencing of DMRs (Fig 2A, dark gray boxes) validated their methylation status obtained from Methyl-Cap-seq experiments and indicated that DAC treatments reduced their level of methylation by half, *in vitro* (Fig 2D and E). Of note, DAC treatment of MDA-MB-231 cells resulted in the upregulation of *UNC5B*, although its promoter was not methylated in any of the cell lines tested (Fig 2B and C), suggesting an indirect regulatory mechanism. However, this upregulation was not observed for the other netrin-1-specific receptors, *UNC5A, UNC5C,* and *DCC* (Fig EV2A and B). Altogether, these data strongly suggest that DNA hypermethylation is involved in the transcriptional silencing of *DAPK1* and *NTN1* in human breast cancer cells.

### Decitabine resensitizes cancer cells to netrin-1 interference *in vitro*

We thus hypothesized that the pharmacological targeting of the DNA methylation machinery may restore functional netrin-1/DR pathways and resensitize netrin-1-low cells to netrin-1 interference. We investigated whether the forced expression of *DAPK1* in the *DAPK1*-negative HMLER cells re-established pro-apoptotic pathways (Fig EV2C and D). As expected, an increase in caspase-3 activity was observed in *DAPK1* transfected cells. Furthermore, this pro-apoptotic effect was partially reversed by adding recombinant netrin-1 (Fig EV2D). We next analyzed cell death by conducting viability assays and caspase-3 activity assays in HMLER cells transfected with *NTN1* siRNA and treated with DAC. While transfections had a mild effect on HMLER cells *per se,* treatment with DAC strongly potentiated the netrin-1 deprivation-induced cell death (Fig EV2F–H).

Keeping in mind the therapeutic perspective of our study, we assessed the effect of combining DAC with a human anti-netrin-1 antibody, net1-mAb (Grandin *et al*, 2016). MDA-MB-231 and HMLER cells were treated with DAC or net1-mAb, or a combination of both drugs. As anticipated, the cell lines were resistant to

net1-mAb alone, while, as previously observed (Lund *et al*, 2011) (Rodriguez-Paredes & Esteller, 2011b), DAC induced cell death in most of the cells tested, as evidenced by DNA fragmentation (Figs 3A and B, and EV2I and J) and viability assays (Fig 3C). The addition of net1-mAb to DAC-treated cells significantly enhanced apoptosis (Fig 3A–C). Moreover, this effect was blocked by the concomitant addition of recombinant netrin-1 (Fig 3A–C), indicating that net1-mAb-induced cell death was specifically linked to netrin-1 neutralization. To confirm that the pro-apoptotic activity observed upon the combined net1-mAb and DAC treatment was not linked to an intrinsic property of DAC, MDA-MB-231 and HMLER cell lines were treated with the 5-azacytidine (Aza), a DNA methylation inhibitor. This treatment resulted in similar gene expression modifications and cell death effects as DAC, when combined with the net1-mAb (Appendix Fig S1). The functional consequences of this combination treatment were also evaluated in additional human breast cancer cell lines. The highest upregulation of both *NTN1* and *DAPK1* was observed when treating cells with 10 µM of DAC (Fig EV3A–E), we thus used this concentration in the combined DAC and net1-mAb treatment to investigate the induction of apoptosis (Fig EV3F). A similar potentiation of cell death by combining DAC and net1-mAb was observed in cell lines (AU565, SKBR3, and MDA-MB-157) exhibiting an upregulation of at least one of 3 genes, *NTN1*, U*NC5B*, or *DAPK1* (Fig EV3C–E).

### The combination of the netrin-1 neutralizing antibody with decitabine inhibits tumor growth and metastasis in animal models

We next investigated whether the inhibition of DNA methylation could also re-establish the netrin-1/DRs-mediated pro-apoptotic pathways *in vivo*. Nude mice were engrafted with MDA-MB-231 cells in the mammary fat pad as an orthotopic site for breast cancer. When tumors were palpable (≈100 mm³), DAC was subcutaneously injected at a therapeutic dose (0.5 mg/kg). Parallel sequencing of amplicons derived from *DAPK1* and *NTN1* CGis indicated that these genes were also hypomethylated (ranging from 5 to 27% depending of the CpG analyzed) in tumors from DAC-treated mice (Fig 3D). Furthermore, we observed that DAC treatments were associated with the re-expression of DAPK1, netrin-1, and UNC5B by the IHC staining of tumor sections (Fig 3E and F).

To evaluate the effect on tumor growth, MDA-MB-231 orthotopic xenografts were performed, and mice were treated by

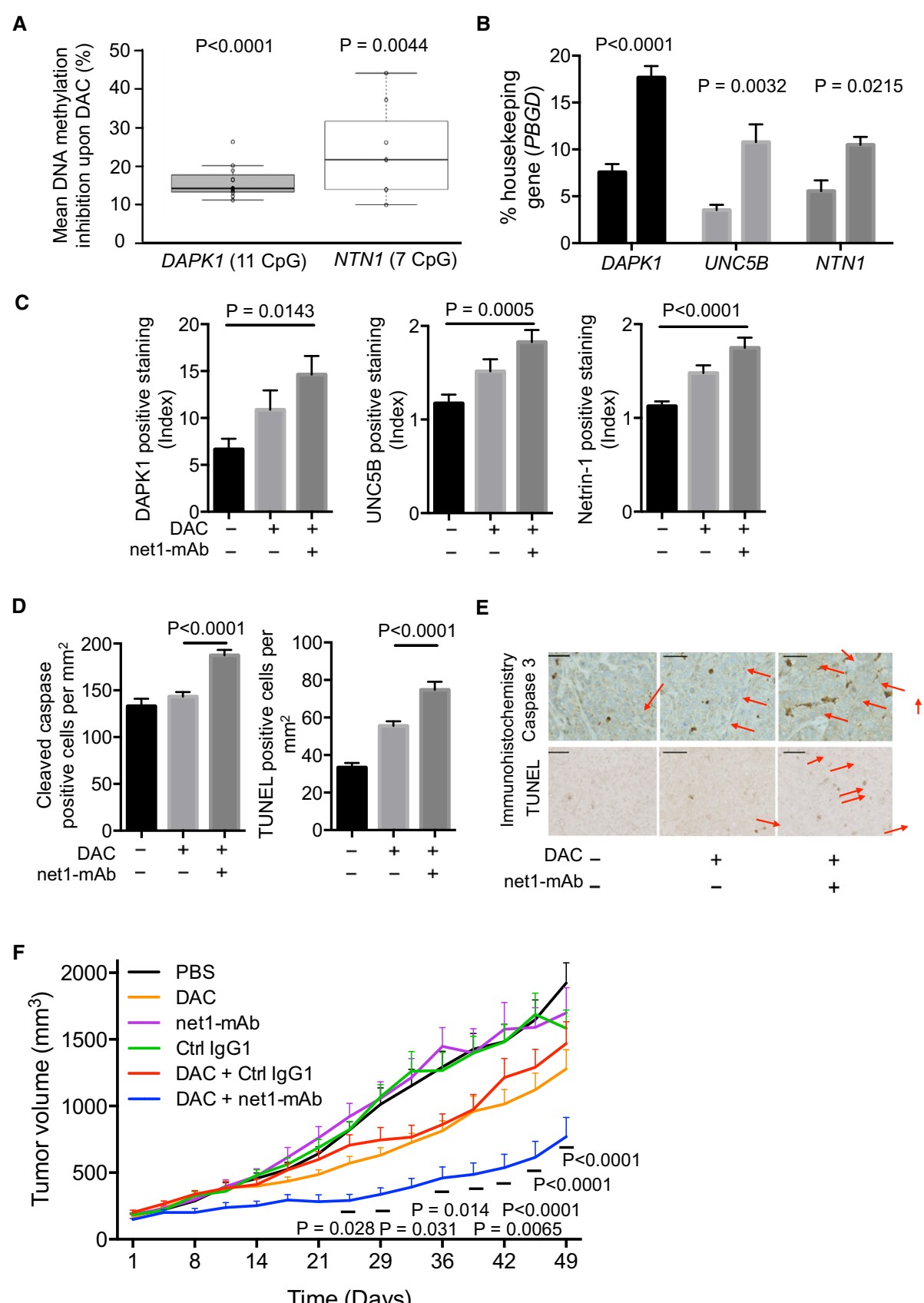

**Figure 4.**

◄ **Figure 4.  DAC treatment triggers upregulation of genes in the netrin-1/dependence receptor signaling pathway, and, when combined with net1-mAb, induces apoptosis and tumor growth inhibition in a mouse model bearing a patient-derived breast tumor.**

A   Loss of DNA methylation of the *NTN1* and *DAPK1* promoters in decitabine-treated PDX tumors, compared with the control PBS group. The percentage of mean DNA methylation of the 11 *DAPK1*-CpGs was 94% (550 amplicons analyzed), while that of the 7 *NTN1*-CpGs was 64% (213 amplicons analyzed). Two-way ANOVA and *post hoc* Tukey test.

B   Expression of *DAPK1*, *UNC5B*, and *NTN1* was measured by qRT–PCR in PDX tumors after 7 days of *in vivo* DAC treatment (0.4 mg/kg). The level of *PBGD* expression was used as an internal control. Data are expressed as mean ± s.e.m. for at least 3 grafts per group. ****$P < 0.0001$, two-way ANOVA and *post hoc* Tukey test.

C   Levels of DAPK1, UNC5B, and netrin-1 were measured in at least 3 independent paraffin embedded xenografts per treatment group by immunohistochemistry. Data are expressed as the percentage of the total tumor surface normalized against the median tumor surface of the PBS group. ****$P < 0.0001$, one-way ANOVA. Error bars = s.e.m.

D   Cleaved caspase-3 and DNA fragmentation (TUNEL) median number of cells per mm$^2$ were measured in treated xenografts. Data represent mean ± s.e.m. from at least 3 tumors per group. ****$P < 0.0001$, one-way ANOVA.

E   Representative sections of the xenograft PDX analyzed in (D). Scale bars = 50 μm. Arrows indicate positive staining.

F   The combination of net1-mAb and DAC reduces human breast tumor growth in immuno-compromised mice. After anesthesia, mice were engrafted in the interscapular area with a ≈ 60 mm$^3$ patient-derived tumor. When the tumors reached 120–150 mm$^3$, mice were injected subcutaneously with decitabine (0.4 mg/kg) or PBS and/or intraperitoneally with net1-mAb (10 mg/kg) or with a human IgG1 control isotype antibody (Ctrl IgG1, 10 mg/kg) from day 1 to day 21. Tumor volumes were measured twice a week; $n = 7$ mice per group. The statistical significance of the differences obtained between the DAC + Ctrl IgG1 group and the DAC + net1-mAb group was determined by two-way ANOVA2 and *post hoc* Tukey test, ****$P < 0.0001$. Error bars = s.e.m.

intraperitoneal injections of net1-mAb, or subcutaneous injections of DAC, or a combination of both treatments for a period of 21 days (Fig 3G and H). While no significant difference was observed between the control groups and mice treated either with net1-mAb or DAC alone, the combination of both drugs strongly inhibited tumor growth (Fig 3G) and led to a reduction in tumor weight (Fig 3H). Furthermore, similar results were obtained using the NET1-mAb antibody, which is scheduled for human trials in 2016 (Grandin *et al*, 2016) (Appendix Fig S2A). Interestingly, the inhibitory effect of the combined treatment on tumor growth was associated with a marked increase in tumor cell death (Appendix Fig S2B and C).

To gain further insights into the mechanisms underlying the tumor suppressive activity of netrin-1 neutralization following DAC treatment, cancer cell proliferation and tumor angiogenesis were analyzed in treated tumors. A small reduction in the proliferative rate (Ki67 staining) was determined in MDA-MB-231 xenografts from mice treated with DAC; however, the differences between treated samples and controls were not statistically significant (Appendix Fig S3A and B). Staining of sections from MDA-MB-231 xenografts with an antibody directed against CD31 (also known as platelet endothelial cell adhesion molecule) revealed that the anti-netrin-1 antibody treatment led to a reduction in angiogenesis. This finding strengthens a previous report, which suggested the implication of netrin-1 during angiogenesis (Bongo & Peng, 2014). However, no statistically significant differences were observed between mice treated with the anti-netrin-1 antibody alone, where no inhibition in tumor growth was detected, and the mice treated with a combination of DAC and net1-mAb, where tumor growth was profoundly affected (Appendix Fig S3A). Hence, the inhibitory effect of the combination of DAC and net1-mAb on tumor growth is likely related to their pro-apoptotic effects, rather than to a change in proliferation or in angiogenesis.

Finally, the effect of this combination treatment on metastasis formation was evaluated in a model previously used for monitoring tumor dissemination (Walker *et al*, 2004; Delloye-Bourgeois *et al*, 2009b). MDA-MB-231 cells previously treated with control IgG1, DAC, and net1-mAb alone or in combination, were seeded onto the chorioallantoic membrane (CAM) of 10 days old chicken embryos. Data obtained indicated that CAMs grafted with cells previously treated with DAC and the net1-mAb exhibited a reduction in the formation of lung metastasis (Appendix Fig S4).

To move toward a model closer to the human pathology, we analyzed the effect of combining DAC and the netrin-1 antibody in mice model bearing a patient-derived tumor (PDX model). Consistently with the data obtained from mice engrafted with cell lines, DAC treatments induced the hypomethylation of *DAPK1* and *NTN1* in our PDX (Fig 4A) and increased the levels of DAPK1, UNC5B, and netrin-1 (Fig 4B and C). Furthermore, IHC staining on tumor sections indicated that the combination treatment increased the apoptotic rate (Fig 4D and E), while it was not correlated with tumor cell proliferation or angiogenesis (Appendix Fig S3C and D). Similarly to the data obtained in MDA-MB-231 engrafted cells, the combined treatment reduced the growth of the PDX tumor when compared to the effects of monotherapies (Fig 4F).

## Tumor growth inhibition induced by combining decitabine with netrin-1 interference is directly mediated by UNC5B and DAPK1

The potential reversibility of epigenetic DNA modifications raised the possibility that inhibition of DNA methylation might induce the re-expression of these genes, and thus, resensitize cells to several pathways resulting in apoptosis. Therefore, we studied gene re-expression following DAC treatment in MDA-MB-231 and HMLER cell lines, by RNAseq. Indeed, the 626 upregulated genes upon DAC treatments (FC untreated/treated cells > 2) both in MDA-MB-231 and HMLER cells were investigated for their Gene Ontology term and their KEGG pathway contributions using the WEB-based GEne SeT AnaLysis Toolkit (Wang *et al*, 2013). Enriched Gene Ontology terms were of various nature and included response to molecules of bacterial origin, response to lipopolysaccharide, biological regulation, cell proliferation, and cytokine activity. Enriched KEGG pathways included KEGG osteoclast differentiation, KEGG MAPK signaling pathway, KEGG rheumatoid arthritis, KEGG cytokine–cytokine receptor interaction, pathways in cancer, and NOD-like receptor signaling pathway (Appendix Fig S5). Genes upregulated upon DAC treatment were associated with many biological functions; however, the profiles of transcriptomic changes suggested that no other pro-apoptotic pathway participated in the synergistic effect of net1-mAb and DAC on cell death observed herein.

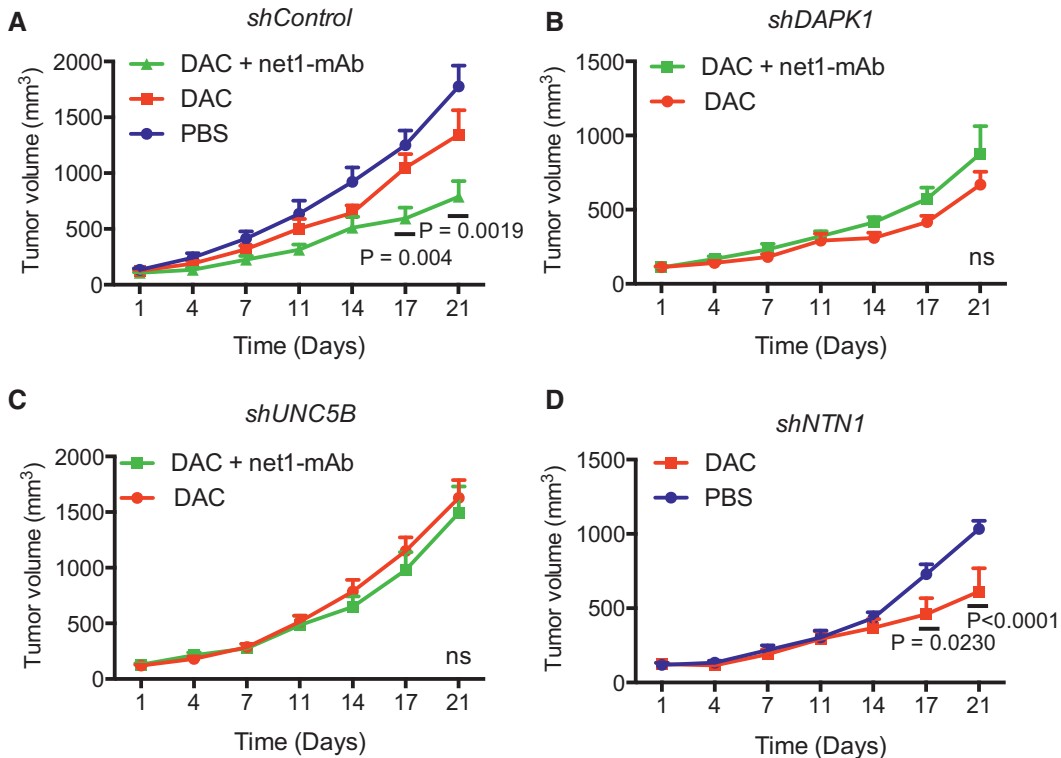

**Figure 5.    Sensitivity of MDA-MB-231 cell lines stably transfected with shRNA targeting *DAPK1*, *UNC5B*, and *NTN1* to treatments combining DAC and net1-mAb.**

A–D    Stably transfected MDA-MB-231 cells bearing a control (A), *DAPK1* (B), *UNC5B* (C), or *NTN1* shRNA (D) were injected into the mammary fat pad of immuno-compromised mice. When tumors reached 100–120 mm³, mice were injected subcutaneously with DAC (0.4 mg/kg) or PBS and/or intraperitoneally with net1-mAb (10 mg/kg). Tumor volumes were measured twice a week, *n* = 8 mice per group. The statistical significance of the differences obtained between DAC group and DAC + net1-mAb group for sh*Control*, DAC group and DAC + net1-mAb group for sh*DAPK1* and sh*UNC5B*, and PBS group and DAC group for sh*NTN1*, respectively, was determined by two-way ANOVA and *post hoc* Tukey test. ****$P < 0.0001$, ns = not significant. Error bars = s.e.m.

It has been reported (Qi *et al*, 2015) that the addition of netrin-1 to the culture medium of human liver cancer, glioblastoma, and embryonic kidney cell lines induces the expression of the Yes-associated protein (YAP), of TAZ, a transcriptional coactivator with a PDZ-binding motif (WWTR1), and of the connective tissue growth factor (CTGF), the transcription of which is initiated by YAP/TAZ. The induction of YAP/TAZ following the addition netrin-1 was correlated with an increase in cell proliferation and following the addition of netrin-1 neutralization by specific anti-bodies, with a decrease in the proliferation and migration of the cell lines analyzed (Qi *et al*, 2015). Therefore, we verified whether, in DAC-treated cells, netrin-1 neutralization leads to the downregulation of the YAP/TAZ signaling pathway, which, in turn, could participate in the anti-tumoral effects associated with the combined treatment. The effect of the net1-mAb on the expression of *YAP, TAZ, and CTGF*, in MDA-MB-231 and HMLER cells treated with DAC, was thus investigated. In MDA-MB-231 cells, DAC treatment increased the level of *TAZ* and *CTGF* expression, while net1-mAb treatment only induced a slight increase in *TAZ* expression. In contrast, in HMLER cells, the addition of net1-mAb only induced the expression of *CTGF* in DAC-treated cells, while the expression of *YAP* and *TAZ* was not significantly modified (Appendix Fig S6A and B). Collectively, these observations did not seem to be in favor of major role of the YAP/TAZ-mediated

signals in the response of cancer cells to the combined DAC/+net1-mAB treatment. In a similar context, *p53* alterations were reported to cooperate with YAP signaling pathways during cancer cell proliferation (Di Agostino *et al*, 2016), raising the question of the implication of TP53 in the apoptotic effect observed. In the panel of cells analyzed, T47D cells exhibited no alteration in *p53* and did not respond to the combined treatment (Fig EV3F), while other cell lines, mutated (MDA-MB-231), blocked (HMLER), or wild type (H460) responded to the combined DAC/+net1-mAb treatment (Grandin *et al*, 2016). Taken together, these data suggested that TP53 does not play a determinant role in the tumor growth inhibitory effect conferred by the combination treatment.

Similarly to the data reported by Roulois *et al* (2015) in colorec-tal cancer cells, DAC treatment of MDA-MB-231 and HMLER cells induced the interferon regulatory factor 7 (IRF7), a key player in the type I interferon (IFN)-dependent immune responses (Appendix Fig S6C and D). We thus investigated the effect of this transcriptional regulator on the expression of *DAPK1, NTN1*, or *UNC5B* upon DAC treatment in MDA-MB-231 and HMLER cells. The transient transfec-tion of these two cell lines with a siRNA targeting *IRF7* strongly reduced *IRF7* expression and prevented its upregulation upon DAC treatment (Appendix Fig S6C and D). However, *IRF7* mRNA deple-tion failed to abrogate DAC induction of *DAPK1, NTN1*, and *UNC5B* upregulation, suggesting that IRF7 and more generally the interferon

(IFN)-like response is not implicated in regulating the expression of these genes following DAC treatment.

Our final working hypothesis was, therefore, that the tumor growth inhibitory effects conferred by the combination treatment were mediated through the pro-apoptotic DR pathway. To validate this hypothesis, we generated MDA-MB-231 cells stably expressing shRNAs targeting *DAPK1*, *NTN1*, or *UNC5B* transcripts. qRT–PCR analyses indicated that the DAC-induced expression of *DAPK1*, *NTN1*, and *UNC5B* was efficiently counteracted by gene silencing in the corresponding MDA-MB-231-shDAPK1, MDA-MB-231-shNTN1, and MDA-MB-231-shUNC5B cells (Fig EV4A). We then investigated the responses of the MDA-MB-231-shRNA cells, when engrafted in mammary fat pad and treated or not with the combination treatment (DAC and the net1-mAb). After validating the levels of DAPK1, UNC5B, and netrin-1 by IHC (Fig EV4B–D), anti-active caspase-3 IHC staining was performed on tumor sections and revealed that the silencing of *DAPK1* or *UNC5B* fully prevents the activation of caspase-3 mediated by the combination treatment, while DAC monotherapy resulted in a marked increase in caspase-3 activity when *NTN1* was silenced. Furthermore, the silencing of *DAPK1* or *UNC5B* mRNA prevented the inhibition of tumor growth induced by the combination treatment, further confirming our working hypothesis (Fig 5A–C). Moreover, *NTN1* silencing was sufficient to induce tumor growth inhibition following DAC monotherapy (Fig 5D). Taken together, these data demonstrate that DAPK1 and UNC5B are the key players involved in the response to the combined DAC and the anti-netrin-1 antibody treatment.

## Discussion

The results presented herein provide a proof of concept that inhibition of DNA methylation can sensitize solid tumors to antibodies mediating tumor cell apoptosis. We previously proposed that in netrin-1-low tumors, inhibition of the netrin-1 dependence receptor (DR) pathway may occur either through the downregulation of the receptors or of key signaling pathway partners in colorectal cancer and neuroblastoma (Bernet *et al*, 2007; Zhu *et al*, 2013). Interestingly, in the panel of breast cancers analyzed in the current study, low levels of *NTN1* expression are associated with the hypermethylation of the CpG island located in the 5′-end of this gene. Furthermore, this correlation between hypermethylation and gene silencing was also found for *DAPK1*, an essential partner in the apoptotic netrin-1/DR pathway. These features were confirmed in breast cancer cell lines, since *NTN1* was downregulated and methylated in HMLER and MDA-MB-231 cells, and *DAPK1* was methylated and downregulated in HMLER cells.

In addition we showed, both *in vitro* and in xenograft models, that combining inhibition of DNA methylation with an anti-netrin-1 antibody resulted in the re-expression of netrin-1 and DAPK1 and led to tumor cell death and tumor growth inhibition. The correlation between *NTN1* and *DAPK1* silencing and DNA hypermethylation does not exclude that their upregulation in DAC-treated cells might be dependent on additional factors. For example, *UNC5B*, while unmethylated, was upregulated in DAC-treated MDA-MB-231 cells, and it was previously shown that in breast cancer cells, the downregulation of some methylated genes is dependent on the deposition

of the methyl-CpG binding domain protein 2 (MBD2) within their promoter sequence (Devailly *et al*, 2015).

It is intriguing that, in the PDX model, the administration of DAC as a monotherapy only resulted in a slight tumor growth inhibitory effect. It is unlikely that the resistance to DAC treatment, used as a monotherapeutic agent, may have resulted from an over- or under-dosage, since several reports have shown that similar DAC doses are efficient in reducing tumor formation in some mouse models, including Apc^Min-induced intestinal neoplasia (Tsai *et al*, 2012), HRAS-G12V-transformed human epithelial kidney, leukemic cells, breast cancer cell lines, and engrafted patient-derived tumors (Mian *et al*, 2011; Kilinc *et al*, 2012). Furthermore, DNA hypomethylation was observed following DAC treatment in MDA-MB-231 xenograft tumors and PDX, indicating that DNA methylation was efficiently inhibited by DAC, *in vivo*. Alternatively, these data would suggest that tumor-growing conditions can overcome, at least partially, the anti-tumor effect of DAC. Experimental approaches (Rodriguez-Paredes & Esteller, 2011b; Yang *et al*, 2012; Azad *et al*, 2013) and clinical trials (ClinicalTrials.gov http://www.clinicaltrials.gov) using combinations of DAC with other epigenetic modifiers and/or cytotoxic agents provide interesting strategies for tumor growth reduction. However, these approaches have overlooked an important aspect of DAC, which is the re-expression of potentially masked targets and pathways. The data reported here provide not only evidence for the importance of ligand/DRs pairs in the regulation of tumor development but also advocate a new strategy based on the specific targeting of genes re-expressed following a DNA hypomethylating treatment. Finally, inhibitors of DNA methylation could "prime" tumors to netrin-1 addiction, and the subsequent treatment with a combination of inhibitors of DNA methylation and an anti-netrin-1 antibody could result in a powerful therapeutic strategy.

## Materials and Methods

### Tumor samples

Human breast cancer samples were provided by the tumor bank of the Centre Léon Bérard (Lyon, France). Fresh tissue samples were obtained during breast surgery, prior to any systemic therapy, snap-frozen in liquid nitrogen, and stored for scientific research in a biological resources repository, according to the French National Ethical Guidelines. Tissue microarrays of paraffin embedded breast tumor sections were obtained from Super Bio Chips (Cliniscience, Nanterre, France).

### Cell lines and treatments

The HMLER, MDA-MB-157, MDA-MB-231, and MDA-MB-231-Luc breast cancer cell lines (Cell Biolabs, San Diego, CA, USA) were maintained in Dulbecco's Minimum Essential Medium F12 Glutamax (DMEM-F12 Glutamax) (Life Technologies). In addition, human EGF 10 ng/ml (Promocell, Heidelberg, Germany), hydrocortisone 0.5 μg/ml, puromycin 0.5 μg/ml (InVitrogen), and insulin 10 μg/ml (InVitrogen) were added to the medium of HMLER cells. AU565 and T47D cell lines were maintained in RPMI medium (Life Technologies), supplemented with insulin 0.2 U/ml (InVitrogen) for T47D

cells. SKBR3 cells were maintained in McCoy's medium (Life Technologies). All of these cell lines were supplemented with 10% FBS (Lonza, Basel, Switzerland) and 1% penicillin/streptomycin (InVitrogen, Carlsbad, CA, USA). Twenty-four hours after plating, cells were grown for 3 days in fresh medium containing various concentrations of decitabine (5-aza-2′-deoxycytidine, DAC) (Sigma-Aldrich) or 2 μM 5-azacytidine (Sigma-Aldrich), renewed every day. The cells were then cultured, under serum deprived conditions for 48 h, in the presence of the net1-mAb anti-netrin-1 antibody (10 μg/ml) (Netris Pharma, Lyon, France), as well as in the presence or not of the recombinant Flag-tagged netrin-1 (5 μg/ml) (Adipogen).

### Enforced gene expression and siRNA experiment

The pcDNA3.1 vector coding for *DAPK1 (Llambi et al, 2005)* was used to enforce the expression of *DAPK1* in HMLER cells. Empty plasmids or plasmids containing a HA-tagged DAPK construct were transfected into HMLER cells using Lipofectamine 2000 (Invitrogen). *NTN1* siRNA has been previously described (Delloye-Bourgeois *et al*, 2009a) and was transfected using Lipofectamine 2000 into HMLER cells, which were previously treated or not with decitabine for 72 h. *IRF7* siRNA was transfected using Lipofectamine 2000, and transfected cells were treated for 72 h with DAC. Scramble siRNA (Sigma) was used as a control.

### Stable shRNA transfection of cell lines

Control, *DAPK1*, *UNC5B,* and *NTN1* shRNA plasmids (Sigma Mission shRNA) were transfected into subconfluent cells using Lipofectamine 2000, according to the manufacturer's protocol. After transfection, puromycin (2 μg/ml) was added to the fresh medium as a selection factor. The selection of transfected cells occurred over a 5-day time-course. Cells were then trypsinized and diluted to obtain 0–2 clones per well in 96-well plates. Cell selection using puromycin (2 μg/ml) was conducted throughout the shRNA transfection, selection, and all of the subsequent experiments, in order to conserve gene downregulation. Following their amplifications, cells were treated or not with DAC, and tested for their level of *DAPK1/ UNC5B/NTN1* expression before and after DAC treatment. Clones that exhibited a "normal" proliferation rate and a low level of expression of the gene of interest were selected.

### Caspase-3 activity and viability assays

Caspase-3 activity was measured as previously described (Llambi *et al*, 2005), (Delloye-Bourgeois *et al*, 2009a) using the Ac-DEVD-AFC substrate assay (Gentaur Biovision, Brussel, Belgium). Alternatively, the percentage of cell death was measured by acridine orange and DAPI staining, using the NucleoCounter NC-3000 system (ChemoMetec A/S, Allerød, Denmark).

### Fluorescent caspase-3 activity

The apoptotic index was measured using the CellPlayer 96-Well Kinetic Caspase 3/7 Apoptosis Kit, according to the manufacturer's protocol (Essen bioscience, Hertfordshire, UK). Three thousands cells were plated in 96-well plates, treated with DAC (10 μM final) or

vehicle (PBS) for 48 h. The cells were then incubated in serum free medium enriched with a kinetic apoptosis reagent (1/5,000e), with net1-mAb (10 μg/ml) and/or with recombinant netrin-1 (5 μg/ml), and with DAC or PBS. Cells were placed in an IncuCyte FLR or ZOOM with a 10 × objective in a standard cell culture incubator at 37°C and 5% $CO_2$ for 48 h. As a marker of proliferation, and to correct for differential proliferation of cells, the total number of DNA-containing objects was counted at the end of the experimental time-course, using Vybrant Green. This number was used to calculate the "apoptotic index", defined as the ratio of the number of caspase-3/7 positive objects to the total number of DNA-containing objects, as recommended by the manufacturer's protocol.

### TUNEL assays

For the detection of DNA fragmentation, cells were cultured on coverslips, and, after treatments, fixed in 4% paraformaldehyde for 20 min. TUNEL assays (terminal deoxynucleodityl-transferase-mediated dUTP-biotin nick end labeling) were then performed using 300 U/ml TUNEL enzyme and 6 μM biotinylated dUTP, according to the manufacturer's guidelines (Roche Diagnostics, Meylan, France).

### Mouse model of xenografts

Five-week-old female athymic Swiss[nu/nu] mice were obtained from Charles River (Ecully, France) and were housed in a specific pathogen-free animal facility. MDA-MB-231 or MDA-MB-231-Luc shRNA cells ($2 \times 10^6$) were resuspended in 200 μl PBS and implanted into the mammary gland fat pad. When tumors reached an approximate volume of 100 mm³, 10 mg/kg of net1-mAb (Netris Pharma, France) or an equal volume of PBS (Life Technologies) was injected intraperitoneally, twice a week for 3 weeks. DAC was then injected subcutaneously into the left flank 3 times a week for 3 weeks (dosage: 0.5 mg/kg; vehicle: PBS) or replaced by PBS in the control groups. Tumor volumes were assessed twice a week with a caliper and calculated with the formula $V = \frac{1}{2}$ (length × width²). Furthermore, a patient-derived xenograft, PDX-HBC-146, was generated and housed in the tumor model laboratory (LMT) of the Centre Léon Bérard. Briefly, following isoflurane anesthesia, mice were engrafted in the interscapular area with a ≈ 60 mm³ patient-derived tumor. When tumors reached 120–150 mm³, mice were injected subcutaneously with DAC (0.5 mg/kg) or PBS 3 times a week for 3 weeks and/or intraperitoneally with net1-mAb (10 mg/kg) or with a human IgG1 control isotype antibody (Ctrl IgG1, 10 mg/kg, Evitria, Switzerland) twice a week for 3 weeks. Concomitantly, in order to evaluate the effects of the treatments on DNA methylation, cell death, cell proliferation, tumor angiogenesis, and protein expression, some xenografted mice were treated for only one week (as described above) before tumor extraction for further analysis. All experiments were performed in accordance with the relevant guidelines and regulations required by the Animal Ethics Committee (accreditation of laboratory animal care by CECCAP, ENS Lyon-PBES).

### Chicken model for tumor cell dissemination

As previously described (Delloye-Bourgeois *et al*, 2009b), MDA-MB-231 cells were initially treated for 3 days with DAC 10 μM or PBS

(vehicle) and then for 2 days with Control Iso-mAb or net1-mAb. After trypsinization, $2 \times 10^6$ cells were suspended in 100 µl of PBS/Matrigel (1:1) and seeded on the chorioallantoic membrane (CAM) of 10-day-old chicks. On day 17, lungs were harvested from the tumor-bearing embryos and genomic DNA was extracted with a NucleoSpin Tissue kit (Macherey-Nagel). Metastasis was quantified by PCR-based detection of the human Alu sequence, using primers for avian repeated element-specific sequences as controls (see Appendix Table S1). For both couples of primers, metastasis was assessed by polymerase activation at 95°C for 2 min followed by 40 cycles at 95°C for 30 s, 63°C for 30 s, and 72°C for 30 s. Genomic DNA extracted from chicken breast and lungs of healthy chick embryos was used to determine the threshold between neuroblastoma (NB) cell-invaded and non-invaded lungs. All experiments were performed in accordance with the relevant guidelines and regulations required by the Animal Ethics Committee (accreditation of laboratory animal care by CECCAP, ENS Lyon-PBES).

### Immunohistochemistry analysis of xenografted cell lines and patient biopsies

Immunohistochemistry staining was performed on an automated immunostainer (Ventana Discovery XT, Roche, Meylan, France) using the DABmap Kit according to the manufacturer's instructions. Tissue samples were fixed in 10% buffered formalin and embedded in paraffin. Following antigen unmasking (citrate buffer pH 7.3, 98°C for 35 min), immunostainings were performed with a polyclonal rat anti-mouse netrin-1 antibody (R&D), a rabbit polyclonal anti-DAPK1 (Acris), or an UNC5B antibody (Sigma), using the Novolink kit (Leica) for revelation. Apoptotic cell staining was performed using a rabbit cleaved caspase-3-specific antibody (Cell Signaling) and the *In Situ* Cell Death Detection kit POD (Roche). Angiogenesis was measured using a rabbit polyclonal CD31 (Platelet Endothelial Cell Adhesion Molecule-1) antibody (ANASPEC), and the rate of cell proliferation was determined using a mouse monoclonal MIB-1-specific antibody (DAKO). Image analysis was performed using a light microscope (Eclipse E400, Nikon France, Champigny, France) equipped with a tri-CDD video camera (Sony, Japan). Quantitative values were determined by morphometric analysis (Histolab, Microvision Instruments, Evry, France) of at least 40 fields per staining at a ×200 magnification for the xenograft tumor sections, and one field per staining of the total tissue section at ×40 magnification for the tissue microarray. The total surface occupied by the tumor was automatically measured, and the positively stained at each surface was expressed as a percentage of the total surface for each field analyzed.

### RNA isolation

Total RNA was extracted from biopsies and xenografts using the TRIzol-Reagent (Ambion, Life Technologies) and from cell lines using the Nucleospin RNAII kit (Macherey-Nagel, Hoerdt, Germany), according to the manufacturer's instructions. For RNA-seq experiments, RNA purity, integrity, and quantification were assessed using agarose gel electrophoresis and a NanoDrop 1000 (Thermo Scientific, Wilmington, DE, USA). Pools of three to five independent extractions were sent for high-throughput sequencing to the Beijing Genomics Institute (BGI, Hong-Kong, China).

### Reverse transcription and quantitative real-time RT–PCR

One microgram of RNA was reverse-transcribed, using the iScript cDNA Synthesis Kit (Bio-Rad, Ivry, France). Quantitative RT–PCR was performed using a Mini opticon (Bio-Rad) and the SYBR super-mix qPCR kit (Bio-Rad). Polymerase was activated at 95°C for 3 min, followed by 45 cycles of amplification and 30 s of cooling. Moreover, gene expression profiles of human samples and cell lines were validated using 3 other standard housekeeping genes, namely *PBGD*, *GAPDH,* and *MBD2*, to confirm the results. Primer sequences for the genes targeted are shown in Appendix Table S1.

### DNA extraction

Biopsy samples and xenografted cell lines were cryogrinded in liquid nitrogen. DNA was then extracted and purified using the Nucleospin tissue DNA extraction kit (Macherey-Nagel), according to the manufacturer's protocol. DNA from cell lines was directly extracted using the standard protocol (Auriol *et al*, 2005).

### Bisulfite treatment of genomic DNA

Two hundred nanograms of genomic DNA was added to 1.8 µg of standard plasmid DNA (pGL3 Basic) and converted using the Epitect Bisulfite kit (Qiagen), according to the manufacturer's protocol.

### Amplification of bisulfite converted DNA and pyrosequencing

Sets of biotinylated *NTN1, DAPK1,* and primers were designed within the promoter region (Fig 1D). As a control, non-modified and modified *GAPDH* sets of primers were used to assess the efficiency of the bisulfite DNA conversion. Modified DNA was amplified in a total volume of 50 µl using the Hotstartaq (Qiagen) kit, in the presence of 1 mM $MgCl_2$ for the *DAPK1* and *NTN1* primers.

Thermal profiles were as follows: 95°C for 10 min followed by 50 cycles of 95°C for 30 s, 50°C (*NTN1*) or 52°C (*DAPK1*) for 30 s and 1 min 30 s (*NTN1* and *DAPK1*) of extension followed by a 10-min final elongation. The primer sequences are shown in Appendix Table S1. The quality and quantity of the PCR product were confirmed on a 2% agarose gel.

PCR products were then pyrosequenced using the Pyromark kit (Qiagen). Reverse single-stranded biotinylated templates were isolated using the PyroMark Vacuum Prep WorkStation (Qiagen). Forty microliters of PCR product was added to 38 µl of binding buffer (Qiagen) and 2 µl streptavidin sepharose high-performance beads (GE Healthcare®). The mixtures were shaken for 10 min at 200 *g* (revolution per minute). After agitation, beads covered with biotinylated DNA were collected and retained on the filter probes by permanent vacuum. The filter probes were successively immerged in different baths: in ethanol 70% for 5 s, in PyroMark denaturation solution for 5 s, and in PyroMark wash buffer 1× for 15 s (Qiagen). The vacuum was then turned off, and the beads fixing DNA strands were released into a 96-well plate containing 25 µl of annealing buffer with 0.3 µM of sequencing primer in each well. The sequencing plate was kept at 80°C for 2 min and at room temperature for 5 min. Pyrosequencing reactions were performed in a PyroMark Q96 system using PyroGold reagents (Qiagen). Results were analyzed using the PyroMark Software.

## DNA methylation analysis

Bisulfite sequencing, used to determine the CpG methylation patterns of *DAPK1* and *NTN1* 5′UTR regions (Appendix Table S1), was performed as described (Beygo *et al*, 2013). Briefly, after a first amplification using sequence-specific primers, PCR fragments were tagged in a second amplification step and sequenced using the Roche/454 GS junior system according to the manufacturer's protocol (Roche emPCR Amplification Method Manual—Lib-A and Roche Sequencing Method Manual). Data were analyzed using Amplikyzer (https://pypi.python.org/pypi/amplikyzer/0.97).

## Methyl-Capture-sequencing (Methyl-Cap-seq)

Genomic DNAs (1 μg) were sheared to an average length of 300–600 bp. Methylated DNA fragments were isolated using beads containing the Methyl-CpG-binding domain of MBD2, according to manufacturer's recommendations (MethylMiner, InVitrogen). After sequencing, using the Illumina 2000 high-throughput sequencing technology by BGI service (Beijing, China), 30–40 million 50-bp reads were obtained for each input and bound fraction and analyzed using R and bioconductor packages.

## RNA-seq analysis

Reads were aligned on the UCSC *Homo sapiens* hg19 genome using TopHat2. Differential expression analyses were performed as described (Kim *et al*, 2013), using the Galaxy server (https://usegalaxy.org/). Only genes with at least 1 read per million (RPM) were kept for subsequent analyses. Enriched Gene Ontology terms and KEGG pathways were identified using Gene Set Enrichment Analysis (Subramanian *et al*, 2005) with genes preranked according to their fold change induced by DAC treatment.

## Data deposition

The MeDP and the RNAseq data from this publication have been submitted to the GEO database the accession number GSE80177.

**Expanded View** for this article is available online.

## Acknowledgements

We are grateful to Drs. A Puisieux and AP Morel for the HMLER cells, and to Sandrine Viala for her nice and efficient work. We thank Dr B. Manship for editing of this manuscript and helpful discussions. This work was supported by institutional grants from the CNRS, the University of Lyon, the Centre Léon Bérard, the Ligue Contre le Cancer, the INCA, the ANR, the ERC, the EC FP7 Hermione-2man and from the Fondation Bettencourt. MG was supported by a fellowship grant from the LabEx DEVweCAN.

## Author contributions

MG contributed to the experimental design and performed the work, with the help of PM, GD, BG, YB, AG, CF, IP, JGD. YB provided some scientific insight and technical support on 454 Junior DNA methylation sequencing experiments and their analysis. NG conducted immunohistochemical staining of xenografted murine samples. AB has provided the net1-mAb antibody and technical support. ZH and AG provided some scientific insight and technical support on pyrosequencing experiments and their analysis. MG participated in the writing

of the manuscript. PM and RD proposed the project, experimental design, and wrote the manuscript.

## Conflict of interest

PM and AB declare having a conflict of interest in this study as co-founders and shareholders of Netris Pharma.

**The paper explained**

**Problem**
In a substantial part of human cancers, netrin-1 (*NTN1*) is upregulated and this upregulation is inhibiting apoptosis induced by its so-called dependence receptors, DCC and UNC5H, and thus promotes tumor progression. However, in other cancers, the selective inhibition of this dependence receptor death pathway relies on the silencing of pro-apoptotic effector proteins. A large fraction of human breast tumors exhibits simultaneous DNA methylation-dependent loss of expression of *NTN1* and of *DAPK1*, a serine threonine kinase known to transduce the netrin-1 dependence receptor pro-apoptotic pathway.

**Results**
Results described in this manuscript propose that the inhibition of DNA methylation by drugs such as decitabine restores the expression of both *NTN1* and *DAPK1*, in netrin-1-low cancer cells. Combination of decitabine with *NTN1* silencing strategies or with an anti-netrin-1 neutralizing antibody potentiates tumor cell death and inhibits tumor growth in different animal models including patient-derived xenografts.

**Impact**
With more than 500,000 death worldwide in 2012, breast cancer is one of the most frequent cancer and represents a therapeutic challenge. Our data suggest that combining DNA methylation inhibitors with netrin-1 neutralizing agents could be a valuable strategy for combating netrin-1 low breast tumors cancer, which may represent 40% of breast cancers.

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
