## [Review Process File · EMBO Molecular Medicine]

Inhibition of DNA methylation promotes breast tumor sensitivity to netrin-1 interference.

Mélodie Grandin, Pauline Mathot, Guillaume Devailly, Yannick Bidet, Akram Ghantous, Clementine Favrot, Benjamin Gibert, Nicolas Gadot, Isabelle Puisieux, Zdenko Herceg, Jean-Guy Delcros, Agnès Bernet, Patrick Mehlen, Robert Dante

Corresponding author: Patrick Mehlen and Robert Dante, CRCL

Review timeline:

Submission date:	14 October 2015
Editorial Decision:	28 December 2015
Revision received:	20 April 2016
Editorial Decision:	10 May 2016
Revision received:	30 May 2016
Accepted:	03 June 2016

Transaction Report:

Editor: Roberto Buccione

1st Editorial Decision

28 December 2015

Thank you for the submission of your manuscript to EMBO Molecular Medicine and again, many apologies for such an exceedingly long delay in reaching a decision on your manuscript.

In this case, as you know, we experienced unusual difficulties in securing three willing and appropriate reviewers. Eventually we received two evaluations, which unfortunately happened to be quiet opposing thus forcing us attempt again to recruit a third reviewer, which we finally succeeded in doing. After some additional delay we finally received a third evaluation, which allows me to finally reach a decision on your work.

Although the reviewers have diverging views on your manuscript, they appear to find your work interesting, although there are many concerns of a fundamental nature. I will not dwell into much detail, and will just highlight a few main points.

Reviewer 1, as you will see, is globally positive. S/he would like you to investigate the effect of the single and combination therapies on metastatic dissemination. The reviewer would also like you to address the apparent pro-metastatic effect of the single treatments. Finally, s/he points to inappropriate statistical analysis and requests additional clarifications.

Reviewer 2 is quite reserved and raises many concerns on your study, including the need to perform experimentation on additional cell lines (see also below). S/he, as does Reviewer 1, is also concerned that statistical treatment is not appropriate. After cross-commenting, one Reviewer noted (and we agree) that many of the issues and some strong statements by this Reviewer can actually be

addressed by simply better explaining the available data and/or making more evident the pertinent statements in your manuscript. Generally speaking, Reviewer 2 does raise several valid points with respect to clarity and quality of data presentation and discussion, and although we do not share his/her globally negative stance, we do encourage you to address them.

Reviewer 3 is positive but also has important concerns. On one hand s/he, as does Reviewer 2, requests additional validation in more cell lines. On the other, the Reviewer notes the unusually high concentration of decitabine used and related to this points, would like to know if an IFN γ response is triggered that could explain the downstream effects. Another well-taken point is that it would be of interest to verify whether YAP signaling is involved in the combination anti-netrin1/decitabine treatment, and whether the latter is p53 status dependent.

We discussed these evaluations and agreed that to address these points would significantly enhance the impact and translational relevance of your study.

As mentioned above, there is a clear need also to improve statistical analysis This is very close to our hearts at EMBO Press and indeed we ask all authors to take direct action on statistics and other related issues upon revision with a mandatory checklist (see further below).

In conclusion, while publication of the paper cannot be published at this stage, we would consider a substantially revised submission, with the understanding that the Reviewers' concerns must be addressed with additional experimental data where appropriate and that acceptance of the manuscript will entail a second round of review.

Please note that it is EMBO Molecular Medicine policy to allow a single round of revision only and that, therefore, acceptance or rejection of the manuscript will depend on the completeness of your responses included in the next, final version of the manuscript.

As you might know, EMBO Molecular Medicine has a "scooping protection" policy, whereby similar findings that are published by others during review or revision are not a criterion for rejection. However, I do ask you to get in touch with us after three months if you have not completed your revision, to update us on the status. Please also contact us as soon as possible if similar work is published elsewhere.

Please note that EMBO Molecular Medicine now requires a complete author checklist (<http://embomolmed.embopress.org/authorguide#editorial3>) to be submitted with all revised manuscripts. Provision of the author checklist is mandatory at revision stage; The checklist is designed to enhance and standardize reporting of key information in research papers and to support reanalysis and repetition of experiments by the community. The list covers key information for figure panels and captions and focuses on statistics, the reporting of reagents, animal models and human subject-derived data, as well as guidance to optimise data accessibility.

I also suggest that you carefully adhere to our guidelines for publication in your next version, including presentation of statistical analyses and our new requirements for supplemental data (see also below) to speed up the pre-acceptance process in case of favourable outcome.

I look forward to seeing a revised form of your manuscript as soon as possible.

***** Reviewer's comments *****

Referee #1 (Remarks):

In the absence of their natural ligand Netrin-1, dependence receptors (DRs), such as DCC, UNC5, and neogenin trigger cell apoptosis, e.g. by signaling via DAPK1. In line, the availability of Netrin-1 (NTN1) in the cancer milieu constrains neoplastic progression and NTN1 is up-regulated in some aggressive cancer histotypes, such as breast cancer, non-small-cell lung cancer, neuroblastoma, and pancreatic adenocarcinoma. In this framework, administration of drugs aimed at impairing NTN1/DR interaction is therapeutically very promising. Inhibition of DNA methylation, e.g. by means of decitabine (DAC), is emerging as an additional therapeutic opportunity to counteract

cancer progression. In their manuscript Grandin and colleagues confirm and extend previous findings on lung cancer cells, showing that DNA methylation represses DAPK1 and NTN1 gene transcription in human breast tumors and DAC treatment efficiently demethylate DAPK1, UNC5B, and NTN1 genes in MDA-MB-231 and HEMLER breast cancer cells, re-sensitizing these cells to NTN1 deprivation-dependent (siRNA or net-1 mAb) apoptosis. Of note, Gene upregulated by DAC participate to several biological functions without any specific enrichment in other pro-apoptotic pathways. Authors also show how, compared to control, DAC alone and net-1 mAb alone, the combination of DAC and net-1 mAb inhibits more efficiently the growth of MDA-MB-231 tumors grown in the mammary fat pad. Importantly, net-1 mAb alone or combined with DAC similarly inhibits tumor angiogenesis, indicating that the observed differences cannot be correlated to inhibition of blood vessel formation. Moreover, stable knockdown by shRNA of either DAPK1, or UNC5B, or NTN1 expression impairs the synergy between DAC and net-1 mAb. Finally, they validate and convincingly confirm their hypothesis in avator mice.

This is a novel, cleanly designed and executed study that provides new relevant information on the use of NTN1 blocking strategies in cancer, which hopefully it may be easily transferred to the bedside; however some issues should still be addressed, explained or commented.

1. Once implanted in the mammary mouse fat pad MDA-MB-231 cells are known to form metastases in distant organs. What are the effects of single and combined therapies with DAC and net-1 mAb on the metastatic spread of MDA-MB-231 and/or HMLER cells?
2. Supplementary Fig. 5. Can authors comment and try to provide experimental evidence or discuss about the fact that when administered alone net-1 mAb and DAC cause and increase in lung metastases compared to control untreated MDA-MB-231-Luc cells? Are these differences statistically significant?
3. Supplementary Fig. 4. Authors should tune down their statement on their PDX mouse ("combined treatment strongly impact on tumor growth"). What the net-1 mAb and DAC combo therapy is causing is a retardation of tumor growth, not stabilization or regression. This trend of growth is clinically defined as a "progression". Statistical analysis on the comparison among growth curves of PBS treated and DAC + net-1 mAb combo treated animals should be made using two-way ANOVA and not by means of Student's t-test on single time points.

Referee #2 (Remarks):

In this paper by Grandin et al, the authors examine the mechanism and consequence of loss of Netrin-1 expression in breast cancer. Having established that Netrin-1 loss of expression occurs by methylation, its receptor UNC5B is not altered, but that its downstream target gene, DAPK is methylated as well, they hypothesize that by using demethylating agent to activate DAPK, the UNC5B mediated apoptosis pathway will be reactivated. In addition. Further, since decitabine will cause reexpression of Netrin-1 (an oncogene) as well, a strategy that combines DAPK re-activation with decitabine and Netrin 1 suppression by antibody based therapy will be an effective strategy to combat these breast cancers. Methylation mediated silencing of Netrin 1 has been reported in other cancers (and in breast cancer in many large methylation databases, including TCGA), but has not been examined in detail as has been done in this paper. DAPK methylation has been known for a few years to occur in multiple types of cancers. Since reactivation of DAPK alone can induce apoptosis, in a situation where Netrin 1 is silent, Netrin expression seems an unwanted side effect that the authors then address. Almost equivalent to combating an iatrogenic event. Nonetheless, effects of the combined therapy are not unimpressive.

Comments:

1. In Figure 1, they provide evidence for loss of Netrin 1 in breast cancer by examining TCGA databases, that in a proportion of tumors, Netrin 1 (43%) and DAPK (23%), the mediator of UNC5B-induced apoptosis are both hypermethylated, while the levels of UNC5B remains unchanged. In a panel of 70 tumors in a microarray, IHC for DAPK, UNC5B and Netrin was

performed. The intensity of staining difference in DAPK-hi vs lo is clear in the sections, while Netrin-hi and Netrin-lo classification is not impressive since Netrin low tumors still show staining, as seen in H and E, and in the same figure in paired samples done in silico from TCGA databases. This is again evident on Fig 1D vs Fig 1G where correlation between expression and methylation is much poorer for Netrin 1 than for DAPK.

Since netrin 1 is the subject of this paper, whether there is loss of Netrin 1 expression and whether it is by methylation is very important. TCGA provides data for 1 or 2 CpGs per gene that have to be validated by an independent assay. Since Netrin methylation in breast cancer is being reported for the first time, sequencing its CpG island in a primary tumor series to support the expression data needs to be performed. In fact the authors did that for (N=not given, but approx. 18-20) tumors. The data is presented as a correlation between methylation and expression in Supp Fig 1D and E, but the exact CpGs that are methylated for each tumor is not presented, so that the density of the CpG island in each tumor is not clear. Also what was the incidence of methylation in Netrin 1 and DAPK in this series? This is the important question - in their tumor series with a different technique, in how many tumors did methylation of a CpG island result in loss of expression by qRT-PCR or RNA seq? This data is presented in Sup1D and E, but is the data for a single CpG, what do the numbers 10,20 etc mean. There is no labeling for the X axis. This independent validation, presented clearly and with conviction, will form the strong foundation for supporting the next series of experiments with cell lines. This data should accompany the main figures in Fig 1 A, not be hidden in the development of experimental models in discussion of Fig 2.

Minor: The Y axis of Figure 1A is not labeled. Same for X-axis of Supp Fig 1 D and E.

2. Two cell lines are selected to set up an experimental model- MDAMB231 and HMLER- in vitro transformed in culture. Since there are 50+ breast cancer cell lines available, the rationale for choosing these two is unclear. For instance, MDAMB231 has a basal level of DAPK that is 5-fold and Netrin level is at least 2 fold above the housekeeping gene, PBGD. HMLER is a SV40 transformed breast cell line. Why no use a cell line derived from breast cancer cell line? Also, curiously, UNC5B in both cell line vacillates with DAC exposure. Although the authors dismiss it as probably due to indirect effects, do the changes affect the subsequent responses? Would an upregulated UNC5B plus reexpressed DAPK have more profound effects compared to just re-expressed DAPK? The opposite effect is seen in HMLER cells. Heterogeneity in responses is not uncommon in cell lines or primary tumors. For this reason, choosing more than just two cell lines and validating the results in a series of at least 4-5 additional cell lines is important. Also choosing cell lines that have undetectable expression of either gene would be better; upregulation of gene expression will then be above 30-40% over baseline. All this cell line methylation and expression data is available in TCGA and in other publications to make the choice of cell lines easier for further confirmation in the lab.

Minor: The error bars in Fig 2B and C are huge. These are cell culture experiments; large error bars are a reflection of poor technique. Replicates need to be redone to acquire more tight values. Correct the statement in page 8 "Upon decitabine treatment ...UNC5B was upregulated in MDAMB231 and not in HMLER" to say that it "...was downregulated (p=0.034) in HMLER".

3. In the next series of experiments, events downstream of induction of DAPK and Netrin are examined, and preclinical testing is performed of the concept that a combination of DAC plus Netrin1 mAb can achieve therapeutic effects. DAC is a global demethylating agent, and one of the worries of such untargeted treatments which have already failed in clinical trials in breast cancer (SU2C breast cancer trials) is that one is unable to determine, in the process of reexpressing silenced genes, how many oncogenic pathways get activated as well, as an untoward/unwanted consequence. The experiments presented here are a classic example, where the oncogenic and TS arms of the Netrin/UNC5B/DAPK pathway get activated. The authors now devise a strategy to combat both events, one of them their own creation due to pharmacological manipulation.

What does the second column in Supplementary Figure S2D indicate? S2I and J were not mentioned in the text. There was no significant induction of apoptosis in S2J as DAC single treatment results in the survival of very few cells (very few DAPI). It would be beneficial to perform additional siDAPK + siNTN + DAC for S2G and S2H, in order to confirm the apoptosis induced by DAC is mediated by DAPK in culture (reversal of cell death is expected).

No error bars in Figure 3A. Why use median? Median is usually used when the spread of points is large and a large number of data points are being taken into consideration. And what statistical test was used for calculating the significance of the median? Is there any representative image for column 5 and 6? Is there significance between 6th and 8th columns in 3A and 3B since the authors claim that additional recombinant netrin-1 can blocked net1-mAb function? The reference of figure

is not clearly indicated (p9, ...netrin-1 mAb (Supplementary Fig S3...). There was no discussion for S3A and S3B in the text. There was no recovery of DAPK1 and very subtle increase in NTN1 upon Aza treatment in MDA-MB-231 cells and these observation apparently do not support the use of the combined therapy in 231 cells.

The sentence "...reestablish the netrin-1 receptor-pro-apoptotic pathway.." does not make sense, since in the presence of Netrin-1, this UNC5B-DAPK-apoptotic pathway is abrogated. Inhibition of methylation by DAC only activates pro-apoptotic pathway in the absence of netrin-1. How was the statistical analysis done in Figure 3C? The inhibition of DNA methylation of DAPK1 and NTN1 was poor as shown in 3C but the authors stated otherwise in the text (p10, ...gene were also hypomethylated). Was the analysis done on an individual CpG of one graft? The error bar was again missing in Figure 3D. What does the fraction of positive staining means? Does it refer to number of cells or area of staining? It is necessary for the authors to present IHC images with higher magnification and larger picture as the subcellular staining of these proteins is unclear.

It is surprising to observe no significant regression of tumor in Net-1 mAB treatment group (Figure 3F) since IHC staining clearly indicated that DAPK1 and netrin-1 are expressed in the graft (Figure 3E left panel). No error bars for S4B and C. The presented image does not support the quantification as DAC treatment apparently resulted in a stronger cleaved Caspase staining. Images of a necessary control- treatment with Net-1 mAB alone was also missing. Error bar was missing in S5A. How was the Ki67 ratio deduced in S5A? Label is missing for S5B. The corresponding main text need to be revised to clearly describe S5A and S5B.

The authors must present the lung-luciferin images for all the treatment groups in S5C; the import of a single lung in a Petri dish is not clear. The x-axis in S5D is misleading as the luminescence was imaged in lung, but not the other organs. How was the number of metastasis quantified? If it was quantified on H/E section, please present the images. Please provide discussion on the increase in lung metastasis upon DAC treatment. Statistical analysis should be evaluated between DAC+net-1 mAB and their respective controls, including DAC alone, Net-1 mAB, and vehicle. The comparison is not with Net1 mAB alone or DAC alone.

4. The authors need to show the percent DNA methylation present in the tumors of the untreated control group in the PDX model (Figure 4A) to demonstrate the effect of DAC on hypomethylation of DAPK and Netrin-1. Analysis of the expression level of DAPK1 and NTN-1 in the PDX model should be shown. Error bar missing in Figure 4B and 4C, and please provide representative IHC images used for the quantification. qRT-PCR of the tissues would be a simple way of showing re-expression of the genes. So much is left to image analysis and statistics in the IHC method, that clean quantification of mRNA or demonstration in a Western blot would be refreshing and confirming. Please provide explanation for using different quantification methods between caspase-3 and the other proteins. Error bar was missing from S6A right panel. IHC staining for shControl xenograft should be performed to support the depletion of each of the tested genes in Figure 5B, C and D.

Referee #3 (Comments on Novelty/Model System):

A large panel of breast cancer cell lines should be employed for validation.

Referee #3 (Remarks):

Grandin et al. propose that decitabine treatment can demethylate NTN1 and DAPK1 in breast cancer cells and resensitize them to netrin 1 suppression or blockade by inducing apoptosis. Although the paper is sufficiently clearly written even for the non-specialist, some stylistic improvements may be considered appropriate at the editorial level (for instance the first two sentences can be rephrased). While this work is original, the medical impact of the combination of decitabine with a NTN1 directed antibody in breast cancer appears too premature at this stage, and the mechanisms of action still incompletely characterized. The in vitro experiments would benefit from validation in additional breast cancer cell lines, that could support more convincingly the conclusions. Specific comments are provided below.

Interrogation of public databases, including the recent TCGA dataset (Cell, 2015) suggests a good

negative correlation between methylation and expression of DAPK1 (as also shown in Suppl Fig 1), but a rather poor correlation for NTN1. This is a fundamental piece of information that should not be placed in supplementary, but might better be shown as a panel in Figure 1. To rule out the confounding role of stromal contamination or different breast histological subtypes, the use of a more extensive panel of breast cancer cell lines can help addressing this specific point.

Prior preclinical and clinical studies have shown superior anticancer activity of low-dose vs high dose decitabine (Cancer Cell 2012; 21(3): 430-446; Cell. 2015 Aug 27;162(5):961-73.) It is unclear why the authors have chosen to employ a rather high concentration (10 microM) and how this correlates with clinically achievable drug levels. Again I would strongly recommend treating with decitabine a larger panel of breast cancer cell lines to evaluate the functional consequences on Netrin 1 signaling.

Decitabine (low dose) treatment has been recently shown to induce a viral mimicry response in colon cancer cells (Cell, 2015). In order to better characterize the molecular basis of decitabine effects, the authors could investigate whether induction of an interferon gamma-like response could trigger transcription factors (for instance MED1) that, in turn, contribute to modulating DAPK1, NTN1 or UNC5B expression.

Recent findings suggest that netrin-1 exerts oncogenic activity through YAP signaling (Proc Natl Acad Sci U S A. Jun 2015;112(23):7255-60). In this regard, it would be relevant to characterize the contribution of YAP signaling modulation on the anticancer effects observed when netrin1 inhibition is coupled with decitabine.

Dependence receptors are involved in p-53 regulated apoptotic pathways. The MDA-MB-231 cell line has previously been reported to express high levels of mutant p53. It would be interesting to investigate whether the apoptotic effects of the combination of decitabine and netrin 1 inhibition are dependent upon p53 status, given the prevalence of TP53 deregulation in breast cancer.

Error bars are missing from some histogram charts.

1st Revision - authors' response

20 April 2016

Editor:

We would like to thank you for giving us the opportunity to submit an amended version of our manuscript "Inhibition of DNA methylation promotes breast tumor sensitivity to netrin-1 interference". We have worked hard to propose a revised manuscript that has included most of the comments of the three referees.

More specifically, the statistical analysis of the data has been revised. Errors are now corrected in the amended manuscript, and errors bars are now SEM. As mentioned by Referee 1, comparison of multiple columns is better performed by ANOVA tests. Therefore we performed, whenever possible, ANOVA tests, using GraphPad Prism, on the data. Although statistic significance of the differences reported in the previous version (using t-test) were similar, ANOVA analysis is now widely used for data comparison.

We also performed additional experiments for evaluating the potential anti-metastatic effect of treatments combining inhibition of DNA methylation and antibodies against netrin-1.

Additional breast cancer cell lines have been used and data obtained indicate that the induction of cell death by netrin-1 antibodies combined with decitabine (DAC) can be also observed, depending on the expression of DAPK1 (effector), NTN1 (ligand), and UNC5B (receptor) expression.

Experiments have also been performed to evaluate the involvement of the YAP signaling pathway, and the genes involved in the response to interferon gamma in the induction of cell death. Data obtained suggested that these two pathways did not play a key role in the response of cell lines to treatments combining DAC and netin-1 antibody.

We have also tried to clarify some points, positions and numbers of CpGs analyzed in DNA methylation study, number of grafts analyzed, methylation level (PDX), and gene expression (Q-RT-PCR on PDX model) upon *in vivo* DAC treatments.

Referee 1

“This is a novel, cleanly designed and executed study that provides new relevant information on the use of NTN1 blocking strategies in cancer, which hopefully it may be easily transferred to the bedside; however some issues should still be addressed, explained or commented.”

We thank the referee for his/her kind and supportive comments. We have now followed his/her suggestions and have performed a large revision to address the key points.

“1. Once implanted in the mammary mouse fat pad MDA-MB-231 cells are known to form metastases in distant organs. What are the effects of single and combined therapies with DAC and net-1 mAb on the metastatic spread of MDA-MB-231 and/or HMLER cells?”

We thank the referee for this important comment. In the model analyzed, we detected metastases in only a minor fraction of mouse engrafted with MDA-MB-231 cells, precluding statistical analysis. Therefore, to better understand the potential effect of the combinatorial DAC and net-1 mAb treatment on metastasis formation we moved to a model previously used for monitoring tumor dissemination (Stupack et al., Nature, 2006, 439:95-9). We grafted MDA-MB-231 cells previously treated with control IgG1, decitabine and/or net1-mAb alone or in combination on the chorioallantoic membrane (CAM) of 10 days old chicken embryos. Seven days after, chicken embryos were then analyzed for metastasis to the lung. Data obtained indicated CAMs grafted with cells previously treated with DAC + net1-mAb exhibited a reduced lung metastasis formation (Appendix Figure S4).

“2. Supplementary Fig. 5. Can authors comment and try to provide experimental evidence or discuss about the fact that when net-1 mAb and DAC cause and increase in lung metastases compared to control untreated MDA-MB-231-Luc cells? Are these differences statistically significant?”

Statistical analysis of the data indicated that the differences observed between mice administered with net-1 mAb and DAC and the control mice were not significant (Student's t-test). In order to better describe these data, we also tried to classify the groups according to the number of mice exhibiting lung metastases versus treatments (see table below), and we observed that the combined treatment reduced the number of mice exhibiting lung metastases. However, statistic analysis of these data did not lead to significant results ($P > 0.05$). Thus, this experiment has been deleted in the revised manuscript, since it does not lead to conclusive results.

	PBS	net1-mAb	DAC	DAC + net1-mAb
Metastasized	60%	78%	57%	22%
Not metastasized	40%	22%	43%	78%
Total number of mice	10	9	7	9

“3. Supplementary Fig. 4. Authors should tune down their statement on their PDX mouse (“combined treatment strongly impact on tumor growth”). What the net-1 mAb and DAC combo therapy is causing is a retardation of tumor growth, not stabilization or regression. This trend of growth is clinically defined as a “progression”. Statistical analysis on the comparison among growth curves of PBS treated and DAC + net-1 mAb combo treated animals should be made using two-way ANOVA and not by means of Student's t-test on single time points””

We agree with this comment, and this sentence was replaced by “Similarly to the data obtained in MDA-MB-231 engrafts, combined treatment reduced tumor growth”. It should be noted that administrated alone DAC impacted PDX-tumor growth, however the combination with the net1-mAb led to tumor retardation appearing earlier (day 21 instead of 32). Furthermore, when compared with the control groups, this effect was statistically significant throughout the period (until day 49) analyzed, in contrast with “DAC-alone-treatment”.

According to this comment on statistical analysis, two-way ANOVA has been performed on the Figures 3G, 4F, 5 and Appendix S2A.

Referee 2:

“1. In Figure 1, they provide evidence for loss of Netrin 1 in breast cancer by examining TCGA databases, that in a proportion of tumors, Netrin 1 (43%) and DAPK (23%), the mediator of UNC5B-induced apoptosis are both hypermethylated, while the levels of UNC5B remains unchanged. In a panel of 70 tumors in a microarray, IHC for DAPK, UNC5B and Netrin was performed. The intensity of staining difference in DAPK-hi vs lo is clear in the sections, while Netrin-hi and Netrin-lo classification is not impressive since Netrin low tumors still show staining, as seen in H and E, and in the same figure in paired samples done in silico from TCGA databases. This is again evident on Fig 1D vs Fig 1G where correlation between expression and methylation is much poorer for Netrin 1 than for DAPK.”

Since netrin 1 is the subject of this paper, whether there is loss of Netrin 1 expression and whether it is by methylation is very important. TCGA provides data for 1 or 2 CpGs per gene that have to be validated by an independent assay.”

“1 a Since Netrin methylation in breast cancer is being reported for the first time, sequencing its CpG island in a primary tumor series to support the expression data needs to be performed. In fact the authors did that for (N=not given, but approx. 18-20) tumors.””

We are very sorry for this error, the number of samples are now given in the legend of the corresponding Fig EV1.

“1-b The data is presented as a correlation between methylation and expression in Supp Fig 1D and E, but the exact CpGs that are methylated for each tumor is not presented, so that the density of the CpG island in each tumor is not clear. Also what was the incidence of methylation in Netrin 1 and DAPK in this series? This is the important question - in their tumor series with a different technique, in how many tumors did methylation of a CpG island result in loss of expression by qRT-PCR or RNAseq?””

DNA methylation data corresponded to two different methods and this was not not described adequately. We apologize for this.

Methylation data of breast tumors were obtained from pyrosequencing, the relative positions of the regions analyzed are shown on Figure 2A, light gray boxes. More precisely, for *DAPK1*, the 3 pyrosequenced CpGs were located at the region +1322 to +1502 from the transcription start site (TSS). The 4 CpG pyrosequenced for *NTN1* were located in the region +2782 to +3037 from TSS.

Methylation data of cell lines were obtained from parallel sequencing of amplicons; the regions analyzed are shown on Figure 2A, dark gray boxes. For *DAPK1*, 11 CpGs, located at nt position -1152 to -989 from the TSS, were analyzed. For *NTN1*, 7 CpG (located at nt position -1478 to -1249 from the TSS) were analyzed.

According to the comment of Referee 2, we individualized the methylation status of each sample and the values (methylation and expression) are represented by a red point for each tumor sample (Fig EV1).

“1-c This data is presented is Sup1D and E, but is the data for a single CpG, what do the numbers 10,20 etc mean. There is no labeling for the X axis. This independent validation, presented clearly and with conviction, will form the strong foundation for supporting the next series of experiments with cell lines. This data should accompany the main figures in Fig 1 A, not be hidden in the development of experimental models in discussion of Fig 2. Minor: The Y axis of Figure 1A is not labeled. Same for X-axis of Supp Fig 1 D and E.”

We apologize for the lack of clarity of these data. As mentioned above, 3 CpG for *DAPK1* and 4 for *NTN1* were analyzed. The X-axis is now labeled (% of DNA methylation) and Y-axis is now labeled. We also took advantage of expand view format to highlight these data (Fig EV1).

“2 Two cell lines are selected to set up an experimental model- MDAMB231 and HMLER- in vitro transformed in culture. Since there are 50+ breast cancer cell lines available, the rationale for choosing these two is unclear. For instance, MDAMB231 has a basal level of DAPK that is 5-fold and Netrin level is at least 2 fold above the housekeeping gene, PBGD. HMLER is a SV40 transformed breast cell line. Why no use a cell line derived from breast cancer cell line? Also, curiously, UNC5B in both cell line vacillates with DAC exposure. Although the authors dismiss it as probably due to indirect effects, do the changes affect the subsequent responses? Would an upregulated UNC5B plus reexpressed DAPK have more profound effects compared to just re-expressed DAPK? The opposite effect is seen in HMLER cells.”

“2-a Heterogeneity in responses in not uncommon in cell lines or primary tumors. For this reason, choosing more than just two cell lines and validating the results in a series of at least 4-5 additional cell lines is important.”

As suggested by Referee 2 and 3, a screen on other cellular models was necessary to ascertain the potential importance of the data shown initially. Thus, we determined gene expression profiles upon DAC treatment, in 5 additional human breast cancer cell lines. Data obtained indicated that combined treatment induced cell death in cell lines re-expressing one or more of the key genes in netrin-1 receptor pathways (*DAPK1*, *NTN1*; and *UNC5B*). These data are shown in the Fig EV3.

“2-b Also choosing cell lines that have undetectable expression of either gene would be better; upregulation of gene expression will then be above 30-40% over baseline. All this cell line methylation and expression data is available in TCGA and in other publications to make the choice of cell lines easier for further confirmation in the lab”

We did an analysis of the databanks: GEO accession number: GSE57342 (DNA methylation analysis) and Expression Atlas - E-MTAB-2706 (expression analysis)). Then we choose cell lines exhibiting various profiles of gene expression for further experiments, Q-RT-PCR and cell death assays (see Figure below and Fig EV 3).

“2-c Minor: The error bars in Fig 2B and C are huge. These are cell culture experiments; large error bars are a reflection of poor technique. Replicates need to be redone to acquire more tight values. Correct the statement in page 8 “Upon decitabine treatment ...UNC5B was upregulated in MDAMB231 and not in HMLER” to say that it “...was downregulated (p=0.034) in HMLER””

We apologize for this problem, the errors bars shown in the previous version were SD; in this amended version we show SEM.

“3. In the next series of experiments, events downstream of induction of DAPK and Netrin are examined, and preclinical testing is performed of the concept that a combination of DAC plus Netrin1 mAB can achieve therapeutic effects. DAC is a global demethylating agent, and one of the worries of such untargeted treatments which have already failed in clinical trials in breast cancer (SU2C breast cancer trials) is that one is unable to determine, in the process of reexpressing silenced genes, how many oncogenic pathways get activated as well, as an untoward/unwanted consequence. The experiments presented here are a classic example, where the oncogenic and TS arms of the Netrin/UNC5B/DAPK pathway get activated. The authors now devise a strategy to combat both events, one of them their own creation due to pharmacological manipulation.”

“3-a What does the second column in Supplementary Figure S2D indicate?”

We apologize for this error, the second column corresponds to “ empty vector + netrin-1”. This is now modified.

“3-a S2I and J were not mentioned in the text”

These two panels were associated with Fig EV2 and are now cited in the main text.

“3-b It would be beneficial to perform additional siDAPK + siNTN + DAC for S2G and S2H, in order to confirm the apoptosis induced by DAC is mediated by DAPK in culture (reversal of cell death is expected).”

This is an interesting suggestion. However double siRNA experiments are frequently associated with a loss of efficiency in terms of gene repression and thus we believe that the gain of information obtained from this challenging would not add much compared to the shRNA for *DAPK1*, *NTN1*, and *UNC5B* performed upon combined treatments, both *in vitro* and *in vivo* (Fig 5 and Fig EV4).

“3-c No error bars in Figure 3A. Why use median? Median is usually used when the spread of points is large and a large number of data points are being taken into consideration. And what statistical test was used for calculating the significance of the median?”

For this analysis, we considered a large number of fields presenting various numbers of cells, and consequently various number of dead cells. In order to determine the physiological trend, the median rate of dead cells appeared to be an appropriate approach. Therefore, we cannot use error bars for the description of these experiments. We used a classical test for multiple comparisons: ANOVA1 test.

“3-c Is there any representative image for column 5 and 6?”

Representative images were shown in Fig S2I and J, unfortunately these data were not cited at the appropriated paragraph. This error is now corrected in amended manuscript.

“3-c Is there significance between 6th and 8th columns in 3A and 3B since the authors claim that additional recombinant netrin-1 can blocked net1-mAb function? The reference of figure is not clearly indicated (p9, ...netrin-1 mAb (Supplementary Fig S3...).”

These columns are now compared with ANOVA1 test (Fig 3A and 3B) and p-values are now shown in the amended version.

“3-d The sentence “...reestablish the netrin-1 receptor-pro-apoptotic pathway..” does not make sense, since in the presence of Netrin-1, this UNC5B-DAPK-apototic pathway is abrogated. Inhibition of methylation by DAC only activates pro-apoptotic pathway in the absence of netrin-1.”

We agree with this comment since this sentence was confusing and was modified as “restore a functional netrin-1 dependence receptors module”.

“3-d How was the statistical analysis done in Figure 3C?”

We have now performed ANOVA2 with Tukey correction, as indicated in the legend of Fig. 2D and 2E.

“3-d The inhibition of DNA methylation of DAPK1 and NTN1 was poor as shown in 3C but the authors stated otherwise in the text (p10, ...gene were also hypomethylated). Was the analysis done on an individual CpG of one graft?”

We have analyzed 11 and 7 CpGs, for *DAPK1* and *NTN1* respectively; these numbers are shown on the new graph (Fig 3C). We agree with the fact that the efficiency of the inhibition of DNA methylation was lower *in vivo* than *in vitro*. This result is not an unexpected result, since the *in vitro* and *in vivo* doses were not the same. However, it has been shown that induction of gene expression in cancer cell lines by DAC can be seen even with a relative low rate of demethylation (Roulois *et al.*, Cell 2015).

“3-d The error bar was again missing in Figure 3D. What does the fraction of positive staining means? Does it refer to number of cells or area of staining?”

This figure has been modified according to this comment.

“3-d It is surprising to observe no significant regression of tumor in Net-1 mAb treatment group (Figure 3F) since IHC staining clearly indicated that DAPK1 and netrin-1 are expressed in the graft (Figure 3E left panel).”

The absence of UNC5B staining may be an explanation for the lack of tumor regression upon net1-mAb single treatment

“3-e No error bars for S4B and C.”

This figure has been modified according to this comment.

“3-e The presented image does not support the quantification as DAC treatment apparently resulted in a stronger cleaved Caspase staining. Images of a necessary control- treatment with Net-1 mAB alone was also missing.”

We agree with the fact that IHC may be not well described, and the labeling points taking in account for quantification are now indicated on the figure by arrows, and control images were also added.

“3-e Error bar was missing in S5A. How was the Ki67 ratio deduced in S5A? Label is missing for S5B. The corresponding main text need to be revised to clearly describe S5A and S5B.”

This figure has been modified according to this comment.

“3-f The authors must present the lung-luciferin images for all the treatment groups in S5C; the import of a single lung in a Petri dish is not clear. The x-axis in S5D is misleading as the luminescence was imaged in lung, but not the other organs. How was the number of metastasis quantified? If it was quantified on H/E section, please present the images. Please provide discussion on the increase in lung metastasis upon DAC treatment. Statistical analysis should be evaluated between DAC+net-1 mAB and their respective controls, including DAC alone, Net-1 mAB, and vehicle. The comparison is not with Net1 mAB alone or DAC alone.”

We thank the referee for this important comment. Statistical analysis of the data indicated that the differences observed between mice administered with net-1 mAb and DAC and the control mice were not significant (Student's t-test). In order to better describe a possible metastatic effect, we also tried to classify the groups according to the number of mice exhibiting lung metastases versus treatments (see table below), and we observed that the combined treatment reduced the number of mice exhibiting lung metastases. However, statistical analysis of these data did not lead to significant results ($P > 0.05$). Thus, this experiment has been deleted in the revised manuscript.

In the model analyzed, we detected metastases in only a minor fraction of mouse engrafted with MDA-MB-231 cells, precluding statistical analysis. Therefore, to better understand the potential effect of the combinatorial DAC and net-1 mAb treatment on metastasis formation we moved to a model previously used for monitoring tumor dissemination in chicken embryo (Stupack et al., Nature, 2008). We grafted MDA-MB-231 cells previously treated with control IgG1, decitabine and/or net1-mAb alone or in combination on the chorioallantoic membrane (CAM) of 10 days old chicken embryos. Seven days after, chicks were then analyzed for metastasis to the lung. Data obtained indicated CAMs grafted with cells previously treated with DAC + net1-mAb exhibited a reduced lung metastasis formation (Appendix Figure S4).

“4-a The authors need to show the percent DNA methylation present in the tumors of the untreated control group in the PDX model (Figure 4A) to demonstrate the effect of DAC on hypomethylation of DAPK and Netrin-1.”

Mean DNA methylation percentage of the 11 DAPK1-CpGs were 94%, 550 amplicons analyzed. Mean DNA methylation percentage of the 7 NTN1 CpGs were 64%, 213 amplicons analyzed. These data are shown in the legend of Figure 4A.

“4-b Analysis of the expression level of DAPK1 and NTN-1 in the PDX model should be shown. Error bar missing in Figure 4B and 4C, and please provide representative IHC images used for the

quantification. qRT-PCR of the tissues would be a simple way of showing re-expression of the genes. So much is left to image analysis and statistics in the IHC method, that clean quantification of mRNA or demonstration in a Western blot would be refreshing and confirming.”

We performed Q-RT-PCR assays of gene expression, these data are now shown in Fig 4.

“4-c Please provide explanation for using different quantification methods between caspase-3 and the other proteins. Error bar was missing from S6A right panel. IHC staining for shControl xenograft should be performed to support the depletion of each of the tested genes in Figure 5B, C and D”

IHC for shControl xenografts were quantitated and statistical significance was assessed by ANOVA1, see below.

Referee 3:

“While this work is original, the medical impact of the combination of decitabine with a NTN1 directed antibody in breast cancer appears too premature at this stage, and the mechanisms of action still incompletely characterized. The in vitro experiments would benefit from validation in additional breast cancer cell lines, that could support more convincingly the conclusions.”

We thank the referee for his/her kind and supportive comments. We have now followed his/her suggestions and have performed a large revision to address these key points.

“Interrogation of public databases, including the recent TCGA dataset (Cell, 2015) suggests a good negative correlation between methylation and expression of DAPK1 (as also shown in Suppl Fig 1), but a rather poor correlation for NTN1. This is a fundamental piece of information that should not be placed in supplementary, but might better be shown as a panel in Figure 1. To rule out the confounding role of stromal contamination or different breast histological subtypes, the use of a more extensive panel of breast cancer cell lines can help addressing this specific point.”

Cell lines analyzed: BT20, EFM19, EFM192A, HCC1187, HCC1419, HCC1500, HCC1569, HCC1954, HCC38, MCF7, MDA231, MDA415, MDA436, MDA453, MDA468, T47D, ZR7530.

Cell lines analyzed: BT20, EFM19, EFM192A, HCC1187, HCC1419, HCC1500, HCC1569, MCF7, MDA231, MDA415, MDA436, MDA453, MDA468, T47D, ZR7530.

Many thanks for this comment, although we cannot exclude a bias introduced by stromal contamination, it should be noted that, in paired samples, hypermethylation was only observed in tumor tissues, the corresponding normal tissues exhibited a very low level of DNA methylation (Fig 1A and 1B). To gain more insight on this point we extracted from databanks (GEO accession number: GSE57342 (DNA methylation analysis) and Expression Atlas - E-MTAB-2706 (expression analysis)) the DNA methylation and the expression level of *DAPK1* and *NTN1* of 17 and 15 cancer breast cell lines, respectively. For *DAPK1*, the methylation level of the 5 CpGs represented on the HumanMethylation450 BeadChip at position -238 to +838 was inversely correlated with the expression level of *DAPK1*, in these cell lines. We also determined a differentially methylated region at *NTN1*-5' end correlating (CpG position +904) with the expression level of *NTN1* (Figure above). Nevertheless, some *NTN1*-low-cell lines were unmethylated suggesting that DNA methylation could be only one of the mechanisms involved in the control *NTN1* expression in these cell lines.

“Prior preclinical and clinical studies have shown superior anticancer activity of low-dose vs high dose decitabine (Cancer Cell 2012; 21(3): 430-446; Cell. 2015 Aug 27;162(5):961-73.) It is unclear why the authors have chosen to employ a rather high concentration (10 microM) and how this correlates with clinically achievable drug levels.”

Preliminary works now cited in the manuscript indicated that high concentrations of decitabine led to high inhibition of DNA methylation and, at concentration used (10 microM), we observed a percentage of inhibition of DNA methylation ranging from 37% to 57%, depending on the gene and the cell line analyzed (Fig EV1D). The *in vitro* experiments were conducted for evaluating the potential mechanistic relationship between DNA methylation and gene expression and their consequences on Netrin-1 signaling. Thus, we chose a relative high dose of decitabine, which was probably more toxic than a low dose, but allowed a very efficient inhibition of DNA methylation. Nevertheless, we agree with the fact that this dose does not correspond to the doses used in therapy, thus, we performed *in vivo* experiments with doses close to the doses used for therapy but analyzed in the tumors from treated animals that inhibition of DNA methylation occurs with the doses used.

“Again I would strongly recommend treating with decitabine a larger panel of breast cancer cell lines to evaluate the functional consequences on Netrin 1 signaling.”

As suggested by Referee 2 and 3, a screen of other cellular models was necessary to ascertain the potential importance of this approach in cell death induction mediated by the netrin-1 receptors pathway. Thus, we determined gene expression profiles upon several concentrations of DAC, in MDA-MB-231, HMLER, and 5 additional human breast cancer cell lines. Data obtained indicated that combined treatment induced cell death in cell lines re-expressing one or more of the key genes in netrin-1 receptor pathways (*DAPK1*, *NTN1*; and *UNC5B*). See Figure EV3

“Decitabine (low dose) treatment has been recently shown to induce a viral mimicry response in colon cancer cells (Cell, 2015). In order to better characterize the molecular basis of decitabine effects, the authors could investigate whether induction of an interferon gamma-like response could trigger transcription factors (for instance MED1) that, in turn, contribute to modulating DAPK1, NTN1 or UNC5B expression.”

We thank the referee for this very interesting suggestion. As reported by Roulois *et al.* (Cell 2015) in colorectal cancer cells, we observed that DAC treatment of MDA-MB-231 and HMLER induced interferon regulatory factor 7 (IRF7), a key player in the type I interferon (IFN)-dependent immune responses. In their elegant paper the authors proposed that silencing IRF7 is inhibiting this type I interferon responses. We thus investigated whether silencing of IRF7 impact on DAC-mediated expression of *DAPK1*, *NTN1* or *UNC5B* upon DAC treatment in MDA-MB-231 and HMLER cells. Transient transfection of these two cell lines with a siRNA targeting *IRF7* strongly reduced *IRF7* expression and prevented its upregulation upon DAC treatment (Appendix Figure S6C and D). However, *IRF7* mRNA depletion failed to modify DAC-induction of *DAPK1*, *NTN1*, and *UNC5B*, suggesting that interferon gamma-like response did not play a major role in the control of the expression of these genes upon DAC treatment, in MDA-MB-231 and HMLER cells (Appendix Fig S6C and D).

“Recent findings suggest that netrin-1 exerts oncogenic activity through YAP signaling (Proc Natl Acad Sci U S A. Jun 2015;112(23):7255-60). In this regard, it would be relevant to characterize the contribution of YAP signaling modulation on the anticancer effects observed when netrin1 inhibition is coupled with decitabine.”

Qi *et al.* (PNAS 2015) have reported that addition of netrin-1 to culture medium of human liver cancer, glioblastoma, and embryonic kidney cell lines induced the expression of Yes-associated protein (YAP), TAZ a transcriptional coactivator with PDZ-binding motif (WWTR1) and the connective tissue growth factor (CTGF), a gene whose transcription is initiated by YAP/TAZ. YAP/TAZ induction upon netrin-1 addition was correlated with elevated cell proliferation and netrin-1 neutralization by specific antibodies with a decrease of proliferation and migration in the cell lines analyzed. Therefore, we cannot exclude that, in DAC treated cells, netrin-1 neutralization resulted in the down regulation of the YAP/TAZ signaling pathway that, in turn, participated to the observed anti-tumor of combined treatment. In order to gain further insights on this point, we investigated the effect of netrin-1 mAb on the expression of YAP, TAZ, and CTGF in MDA-MB-231 and HMLER cells treated with DAC. In MDA-MB-231 cells, while DAC elevated TAZ and CTGF expression, a small increase of TAZ expression upon net1-mAb addition was only observed in DAC treated cells. In contrast, in HMLER cells, addition of netrin-1 antibodies induced only CTGF expression in DAC treated cells, while the expression of YAP and TAZ was not significantly modified (Appendix Fig S6A and B). Taken together, these data suggested that YAP signaling pathway did not play a key role in the anticancer effect of treatments combining DAC and net1-mAb.

“It would be interesting to investigate whether the apoptotic effects of the combination of decitabine and netrin 1 inhibition are dependent upon p53 status, given the prevalence of TP53 deregulation in breast cancer.”

In the panel analyzed (Fig EV3F), T47D cells exhibited no alteration of p53 and appears not to respond to combined treatment (Fig EV3F). Moreover, we failed to see any difference between p53 mutant (MDA-MB-231), p53 "dead" (HMLER) or p53 wild-type (H460) upon cell death induced by combining DAC and net1-mAb, suggesting that p53 is not a key player in the apoptotic effect observed.

We are grateful to the reviewers for their comments, which we believe have strengthened the manuscript. We believe that this manuscript provides new insights in the fields of oncology and apoptosis.

If we should send additional information, please let me know. We thank you in advance for your consideration of the revised manuscript.

2nd Editorial Decision

10 May 2016

Thank you for the submission of your revised manuscript to EMBO Molecular Medicine. We have now received the enclosed reports from the referees that were asked to re-assess it. As you will see the reviewers are now globally supportive and I am pleased to inform you that we will be able to accept your manuscript pending the following final amendments:

1) Reviewer notes that the manuscript could benefit from some work on English usage and I agree. Unfortunately we cannot provide comprehensive language editing and therefore suggest that you make an effort to revise the manuscript in this respect. In the meanwhile, please see below an edited version of the abstract for your approval.

2) As per our Author Guidelines, the description of all reported data that includes statistical testing must state the name of the statistical test used to generate error bars and P values, the number (n) of independent experiments underlying each data point (not replicate measures of one sample), and the actual P value for each test (not merely 'significant' or ' $P < 0.05$ ').

3) We encourage the publication of source data, particularly for electrophoretic gels and blots, with the aim of making primary data more accessible and transparent to the reader. Would you be willing to provide a PDF file per figure that contains the original, uncropped and unprocessed scans of all or at least the key gels used in the manuscript? The PDF files should be labeled with the appropriate figure/panel number, and should have molecular weight markers; further annotation may be useful but is not essential. The PDF files will be published online with the article as supplementary "Source Data" files. If you have any questions regarding this just contact me.

4) I note that there is a reference to "Supplementary Table S1" on p. 28. This should be changed to "Appendix Table 1".

Please submit your revised manuscript within two weeks. I look forward to seeing a revised form of your manuscript as soon as possible.

***** Reviewer's comments *****

Referee #1 (Remarks):

The authors have nicely and convincingly addressed all the points raised. I am fully satisfied with the revisions. This is an excellent paper.

Referee #2 (Comments on Novelty/Model System):

The authors have taken pains to correct most of the problems detailed in the last report

Referee #2 (Remarks):

The authors have painstakingly revised the manuscript. English language is problematic throughout, so extensive editorial revision is recommended.

Referee #3 (Comments on Novelty/Model System):

The authors have satisfactorily addressed my concerns by including additional breast cancer cell lines. The discussion has also improved.

2nd Revision - authors' response

30 May 2016

Thank you for your kind comments on our manuscript, "Inhibition of DNA methylation promotes breast tumor sensitivity to netrin-1 interference" by Grandin et al. As suggested we have asked in depth correction by an English native speaker, we have improved the statistical aspect of each of the figures and have changed the "Supplementary Table S1" into Appendix Table 1.

Thank you very much for your continued support.

Corresponding Author Name: P. Mehlen and R. Dante

Journal Submitted to: Embo Molecular Medicine

Manuscript Number: EMM-2015-05945